# Potential effects of metal ion induced two-state allostery on the regulatory mechanism of *add* adenine riboswitch

Lei Bao [1✉], Wen-Bin Kang[1] & Yi Xiao[2]

Riboswitches normally regulate gene expression through structural changes in response to the specific binding of cellular metabolites or metal ions. Taking *add* adenine riboswitch as an example, we explore the influences of metal ions (especially for $K^+$ and $Mg^{2+}$ ions) on the structure and dynamics of riboswitch aptamer (with and without ligand) by using molecular dynamic (MD) simulations. Our results show that a two-state transition marked by the structural deformation at the connection of J12 and P1 ($C_{J12-P1}$) is not only related to the binding of cognate ligands, but also strongly coupled with the change of metal ion environments. Moreover, the deformation of the structure at $C_{J12-P1}$ can be transmitted to P1 directly connected to the expression platform in multiple ways, which will affect the structure and stability of P1 to varying degrees, and finally change the regulation state of this riboswitch.

[1] School of Public Health, Hubei University of Medicine, Shiyan, Hubei 442000, China. [2] Institute of Biophysics, School of Physics, Huazhong University of Science and Technology, Wuhan, Hubei 430074, China. ✉email: bolly@whu.edu.cn

Riboswitches are complex folding domains located in the 5' un-translated regions of messenger RNA in prokaryotes[1–4]. They usually regulate gene expression through the structural changes triggered by the specific binding of their cognate metabolites. A typical riboswitch is composed of two distinct but interacting parts: the aptamer domain (AD) is responsible for recognizing and capturing the ligands, whereas the expression platform (EP) executes gene regulation in response to the ligand binding event in AD[3]. Therefore, to elucidate the nature of conformational transition in different environments is the primary task to understand its regulation mechanism.

As regulatory elements based on RNA structures, riboswitches have a high density of negative charges on the phosphate backbone. Thus, ligand binding alone cannot bring riboswitches into the final regulatory domains, and metal ions must play an important role in the corresponding structural transitions[5,6]. Metal ions can not only compensate the negative charges of riboswitches by dissociating around them in a continuous way, but also change local structures of riboswitches through special binding[7–12]. In the past decades, many experiments have been carried out to study the influences of metal ions on the structures of riboswitches and related regulatory mechanisms, and several insights are obtained as follows: (i) due to the strong electronegativity or structural specificity of ADs, some riboswitches may directly use metal ions (e.g. $Mn^{2+}$ or $Mg^{2+}$) as their cognate ligands[13–15]; (ii) the ligands of some riboswitches have a nonuniform charge distribution or are negatively charged (e.g., lysine, glycine or fluoride ion), so their electrostatic properties need to be balanced by interaction with metal ions for better binding to the corresponding ADs[16–18]; (iii) metal ions are critical for the prefolding of riboswitch AD and subsequent ligand recognition[19–23]. For example, the addition of $Mg^{2+}$ ions can partially pre-organize the conformationally heterogeneous adenine-sensing riboswitch AD in ligand-free state to facilitate ligand binding[19]; (iv) during the regulation process of riboswitch, the complex structural interplay between AD and EP is not only ligand-directed but also strongly coupled to the presence of metal ions[24,25]. Especially, the favorable charge/size ratio of $Mg^{2+}$ ion allows it to shift the ligand-mediated folding pathways of some riboswitches from "induced-fit" (binding first) to "conformational selection" (folding first)[26,27].

The purine riboswitches, as a famous family of structurally simple riboswitches, still represent a variety of mechanisms employed by more complex riboswitches. This family of riboswitches usually has similar AD structure (containing P1, P2, and P3 stems) centered upon a three-way junction (taking adenosine deaminase (*add*) adenine riboswitch[28,29] as an example, see Fig. 1a, b), but controls gene expression in multiple ways due to the differences in sequences of P1 and EP. So far, existed experiments mainly focused on the pre-folding process before ligand recognition and the effect of ligand binding on the relevant regulatory mechanisms of purine riboswitches. Firstly, a hierarchical pre-folding of AD prepares for the subsequent ligand recognition. In general, the two stems P2 and P3, as well as the interactions between them by forming kissing loops L2-L3 are almost completely developed even before ligand binding[30–32]. Afterward, the capture of ligands by the flexible junction region (J12, J23, and J31) and the stabilization of P1 are particularly critical for the downstream EP of purine riboswitch to fulfill its regulatory function. Notably, the AD of purine riboswitch would appear various flexible intermediate states during the process of structural transition induced by ligand recognition[33–40]. These intermediate states are difficult to observe as a result of their high flexibility and sensitivity to the surrounding environment. However, they are of great value in refining the understanding of regulation mechanisms of riboswitches, and experiments are therefore dedicated to their exploration with improved techniques. Although the structural changes in purine riboswitch and some of the important intermediate states that accompany this process have been studied experimentally to some extent, the specific roles of metal ions in these issues are still poorly understood, mainly due to the following limitations of current experimental techniques: (i) experiments based on X-ray diffraction only yield transient structures of individual intermediate states of the riboswitches, but cannot describe dynamic details of the conformational transition processes involved; (ii) while NMR

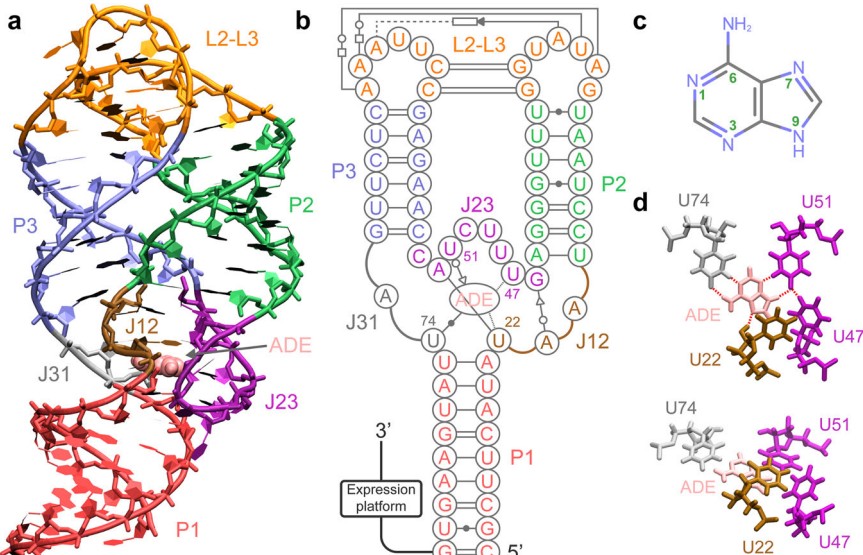

**Fig. 1 The structure of *add* adenine riboswitch and the geometry of ligand binding pocket. a** The tertiary structure of *add* adenine riboswitch in a complex with the adenine (ADE) ligand generated using VMD. The adenine ligand is depicted with a van der Waals representation, and each part of riboswitch is marked with different colors. **b** The secondary structure of *add* adenine riboswitch, which is marked similarly to the tertiary structure. The tertiary interactions are indicated with the notation of the Leontis–Westhof–Zirbel method[87]. For the purpose of clarity, several unimportant tertiary contacts are not shown. **c** Chemical structure of adenine molecule (nitrogenous groups are identified in light blue). **d** The top and top-side views of ligand binding pocket. The hydrogen bonds are indicated by red dashed lines (top view), and the base stacking information in the binding pocket is shown in top-side view.

and single-molecule FRET can be used to study the dynamics of biomolecules in solution and indirectly speculate on the effect of metal ions on them[36,39], neither can give direct evidence of the roles of metal ions in the successive conformational transitions that occur in riboswitches[40]; (iii) even in X-ray crystallography, it is quite difficult to identify $Mg^{2+}$ ions (most effective for stabilizing the native RNA structure) by checking residual electron density maps because they have the same number of electrons as $Na^+$ ions and water molecules[41]; (iv) the contribution of monovalent metal ions (such as $K^+$ and $Na^+$) in the free or weakly bound state to the stability of the RNA structure cannot be ignored either[12,42], but almost all experimental techniques are unable to assess it in detail.

With the rapid development of computer technology and the continuous improvement of molecular force fields, molecular dynamics (MD) simulations are effective in providing direct and detailed understanding of the microscopic mechanisms underlying experimental phenomena. To date, most MD-based research efforts focus on probing the ligand binding[43–46] and the subsequent effect on the structural stability of purine riboswitch[47–50], while others have discussed the mechanisms of long-range interactions in this family of riboswitches[51,52]. However, a detailed understanding of interaction modes between purine riboswitches and metal ions at atomic level is still lacking. The limitations of previous work may come from several aspects, including the sampling time, the accuracy of molecular force field, the way of analysis, and so on.

To obtain a comprehensive knowledge for the effect of metal ions on the structural transition mechanism of purine riboswitch, we systematically investigated the dynamics of *add* adenine riboswitch, a representative member of this family, in several typical ionic environments. This riboswitch traps adenine ligand (Fig. 1c) in the pocket formed by a three-way junction (Fig. 1d) and adopts the corresponding structural transitions to regulate the expression of *add* gene in *Vibrio vulnificus*[28]. Through specially designed analytical perspectives, we obtained several important structural forms of *add* adenine riboswitch aptamer (AARA) in multiple ionic environments and determined the free energy differences between them. Our results demonstrate that ligand binding pocket is the most sensitive region to metal ions in the entire AARA structure. Metal ions (in particular $Mg^{2+}$) distort the phosphate backbone near the binding pocket, forcing the interaction patterns to change simultaneously with the three-dimensional structure around this region, and further causing remarkable effects on the ligand binding affinity and structure of P1 stem, which may eventually lead to a shift in the regulatory state of *add* adenine riboswitch.

## Results

### The influence of metal ions on the global structure of AARA.
The calculations of $r_{\Delta d}(P_i, P_j)$ clearly show several regions of AARA have undergone great structural changes at different ionic conditions (Fig. 2). In $K_{neu}$_free system, an increase in distance (~40% compared with crystal structure) is found at the connection between J12 and one strand (5' end) of P1 (simply marked as $C_{J12-P1}$), which also directly leads to the increase of the distance between J31 and this strand. On the other hand, compared with the crystal structure captured in the experiment, the whole structure of AARA is somewhat more compact in $Mg_{0.3}$_free system than that in the crystal structure (show more dark blue areas in $r_{\Delta d}(P_i, P_j)$ heat maps). Additionally, for $Mg_{0.3}$_free system, several aspects are still worth our concern: (i) the groove formed by two strands (from P2 and P3, respectively) connected to J23 are somewhat narrowed; (ii) compared with ligand-bound form, the residue U48 on J23 is far away from some residues

(especially for U20) on P1, which may make it more solvent-exposed.

Interestingly, for $K_{0.3}$_free and $K_{neu}$_bound systems, the results from two independent simulations of them both show different properties. The structural change of AARA in $K_{0.3}$_free_1 and $K_{neu}$_bound_1 are similar to those in $K_{neu}$_free system, such as the elongated $C_{J12-P1}$. However, the dynamics of AARA in $K_{0.3}$_free_2 and $K_{neu}$_bound_2 are different from both $K_{neu}$_free and $Mg_{0.3}$_free systems. In these two situations, the elongation of $C_{J12-P1}$ is not as obvious as those in $K_{neu}$_free system (~20% for $K_{0.3}$_free_2 and ~10% for $K_{neu}$_bound_2). To clearly examine the detailed changes of this region, the distance between residues U20 and A23 ($d_{U20-A23}$) versus simulation time were calculated for all eight simulation trajectories (Supplementary Fig. S3). In $K_{neu}$_free and $Mg_{0.3}$_free systems, this distance is quite stable, which are ~12 Å for $K_{neu}$_free and ~8.5 Å for $Mg_{0.3}$_free, respectively. Similar to the $K_{neu}$_free system, $d_{U20-A23}$ maintains ~12 Å in the whole simulation processes of $K_{0.3}$_free_1 and $K_{neu}$_bound_1. Nevertheless, the stabilities of $d_{U20-A23}$ are not strong in $K_{0.3}$_free_2 and $K_{neu}$_bound_2, and it is found to undergo transitions between the values of 8.5 Å and 12 Å over time. The heat map of standard deviation of $r_{\Delta d}(P_i, P_j)$ strongly confirms the above facts. (Supplementary Fig. S4). Therefore, exploring the structural changes of $C_{J12-P1}$ region may be the key to understand the effect of metal ions on the dynamics of AARA.

To intuitively observe the global structure characteristics of AARA in different ionic environment, the structure clusters of AARA and specific binding sites (occupancy >0.1 Å$^{-3}$ for the whole simulation process) of metal ions around it for each whole trajectories were extracted (Fig. 2 and Supplementary Fig. S5). Indeed, as previously discussed, the largest structural difference caused by the change of ionic environment occurs in region $C_{J12-P1}$, and the structure of stem P1 directly connected with this region is also greatly changed. Besides, compared with other systems, a considerable narrowing of the major groove formed between stems P1 and P3 is found in $Mg_{0.3}$_free system. Our results also indicate that the distribution of strong binding sites for $K^+$ ions is not sensitive to the ion concentration, and is mainly located in the grooves formed by stems P2 and P3. In $Mg_{0.3}$_free system, some strong binding sites are located in the groove formed by P1 and J31, which is not found in the systems containing only $K^+$ ions. In the absence of $Mg^{2+}$ ions, P1 and J23 are close to each other, accompanied by the aggregation of $K^+$ ions around this region, which may also have an important effect on the structure of ligand binding pocket. In brief, the structure of AARA may be greatly correlated to the type and concentration of metal ions. In particular, AARA shows strong stability in $Mg^{2+}$ ionic solution even without ligand binding.

### The free energy landscape of AARA in different ionic environments.
To quantify the characteristics of structural transformation, umbrella sampling simulations were performed to determine relative stability of AARA in solution. Base on the analysis from last section, the configuration of $C_{J12-P1}$ can well reflect the structural differences of AARA at different ionic conditions. Therefore, we chose $d_{U20-A23}$ (closely related to structure of $C_{J12-P1}$) as the reaction coordinate (Fig. 3a). Then, we computed the potentials of mean force (PMF) along the one-dimensional $d_{U20-A23}$ coordinate space for all four simulation systems (Fig. 3b). The results of PMF show that only one single potential well is found in $K_{neu}$_free system or $Mg_{0.3}$_free system, with a minimum at ~12 Å for $K_{neu}$_free system or ~8.5 Å for $Mg_{0.3}$_free system. In case of $K_{neu}$_free system, the difference of free energy between the minimum (~12 Å) and position of ~8.5 Å is ~−1.8 kcal·mol$^{-1}$, and this difference for case of $Mg_{0.3}$_free

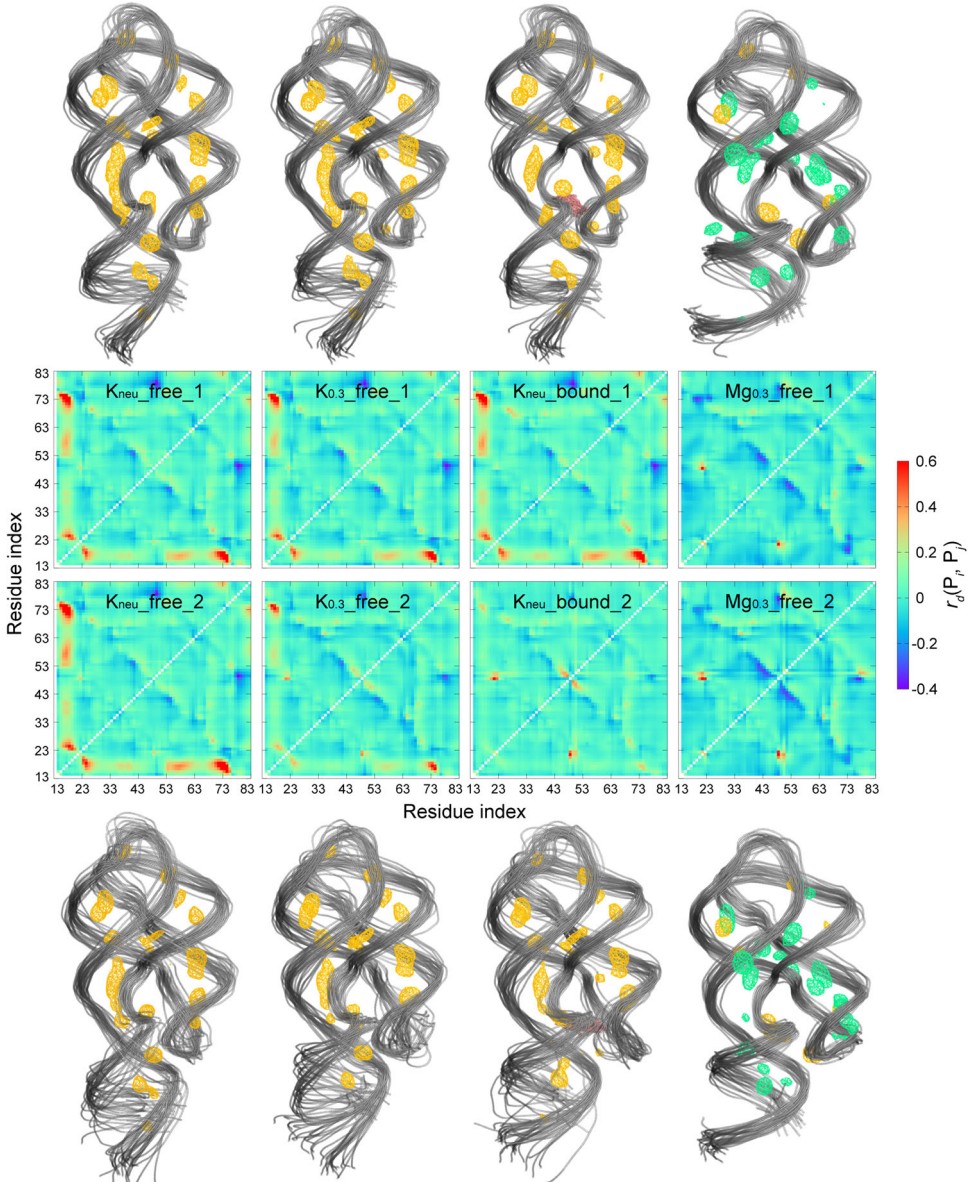

**Fig. 2 The structural features of AARA in different ionic environments and the metal ion distribution around it.** The change ratio $r_{\Delta d}(P_i, P_j)$ of the average distance between every two phosphate groups relative to that in crystal structure for all eight simulation trajectories. The color bar shows the variations in $r_{\Delta d}(P_i, P_j)$ from −0.4 (dark blue) to 0.6 (red). The structure clusters of AARA and specific binding sites of metal ions around it in all eight simulation trajectories (all of 1.2 μs data). Here, the sampling interval for generating structure clusters is 40 ns (30 conformations in total). The specific binding sites are described by the spatial occupancies of metal ions with an isosurface of 0.1 Å$^{-3}$ (K$^+$ in orange and Mg$^{2+}$ in green). The time-averaged spatial occupancies of metal ions were converted into density maps using the VolMap tool of the VMD molecular visualization program, and we chose a resolution of 0.5 Å$^3$.

system is ~−2.5 kcal·mol$^{-1}$, which means that only one stable conformation can exist in K$_{neu}$_free or Mg$_{0.3}$_free system. Unexpectedly, both K$_{0.3}$_free system and K$_{neu}$_bound system contain two potential wells with minima at ~8.5 Å and ~12 Å. However, the peak lying between these two wells is not high (~0.9 kcal·mol$^{-1}$ for K$_{0.3}$_free and ~0.7 kcal·mol$^{-1}$ for K$_{neu}$_bound), which allows AARA structure to change easily between the two typical conformations without paying too much energy cost. The results fully verify our previous analyses of structure clusters of AARA, and clearly indicate the discrepancy of free energy between two typical structures of AARA at different ionic conditions. For all four systems, distances below 8 Å led to unphysical spatial exclusion, resulting in sharp rises of free energies for these distances.

**Comparison of AARA structural properties between K$_{neu}$_free system and Mg$_{0.3}$_free system.** Obviously, in the presence of Mg$^{2+}$ ions, C$_{J12-P1}$ is more compact than that without Mg$^{2+}$ (especially for K$_{neu}$_free system) (Fig. 4a). Previous studies indicate that the influence of high valence metal ions on the structure of nucleic acids mainly comes from the specific binding, and Mg$^{2+}$ ions are unusually efficient[7,9,42]. To accurately locate the specific binding sites of metal ions on AARA structure and find the related residues around them, the heat maps for number of common metal ions in the range of 6 Å around every two phosphate groups were computed (Fig. 4d). Results show that once Mg$^{2+}$ ions exist in the solution, it is difficult for K$^+$ ions to be stably bound to AARA. Compared with K$^+$ ions, Mg$^{2+}$ ions prefer to be bound in the region adjacent to C$_{J12-P1}$ and J31.

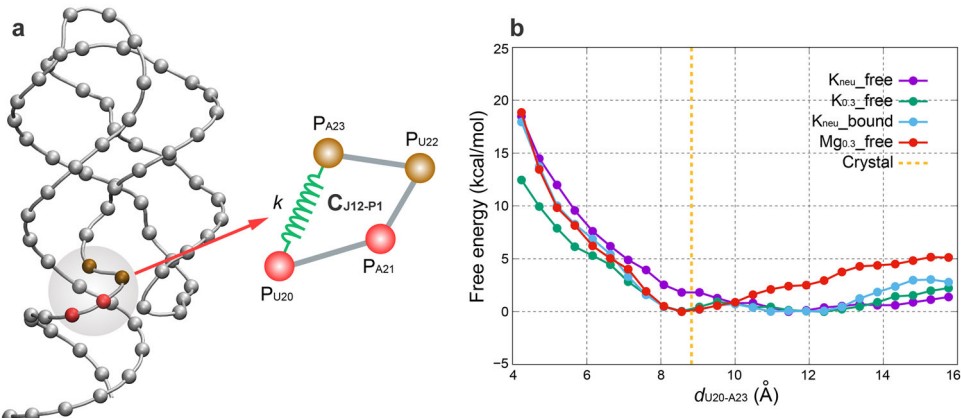

**Fig. 3 The free energy landscape of AARA in different ionic environments. a** The schematic diagram of reaction coordinate selection. The distance between residues U20 and A23 was chosen as the reaction coordinate, and constraints with a harmonic force constant of 5 kcal·mol$^{-1}$·Å$^{-2}$ was applied between them. **b** Potential of mean force from umbrella sampling simulations showing the relative free-energy landscapes of all four systems along the $d_{U20-A23}$ coordinate space, and the orange dashed line corresponds to $d_{U20-A23}$ in the crystal structure (~8.8 Å). The error bars calculated through Monte Carlo bootstrapping is smaller than the size of data point and cannot be shown in the figure.

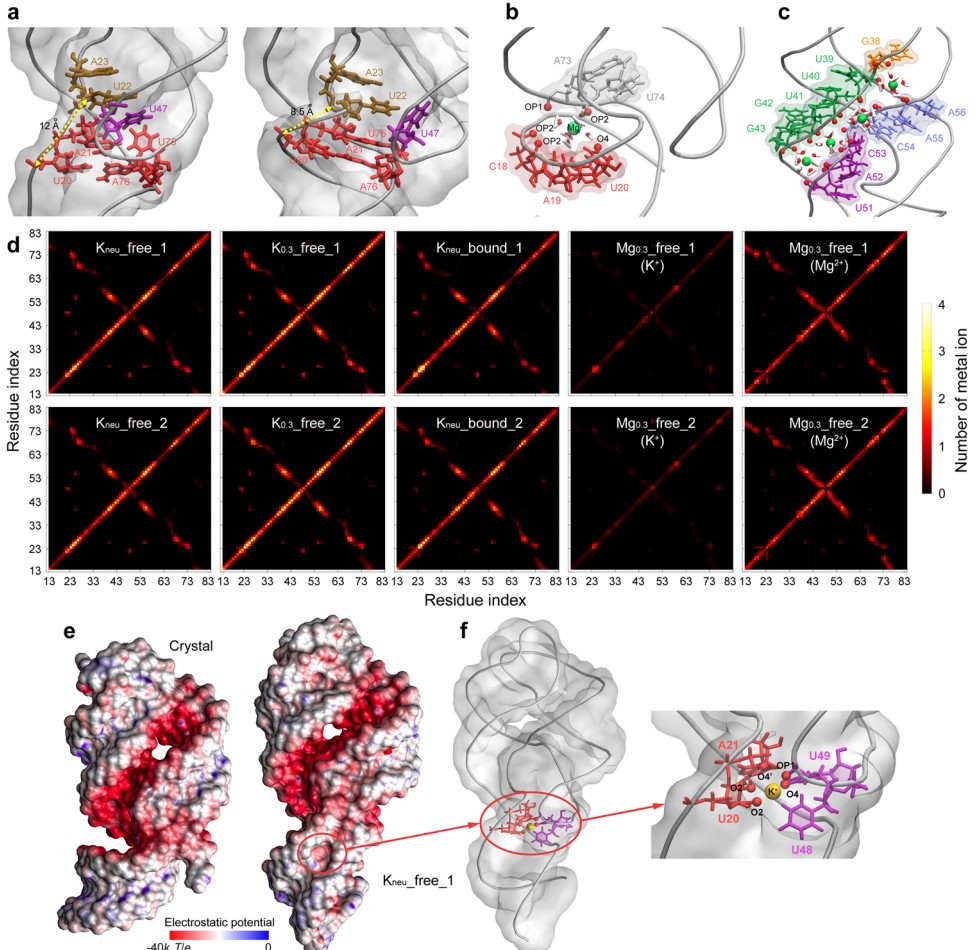

**Fig. 4 Details of the interaction pattern between metal ions and AARA structure. a** Snapshots of the structure around ligand binding pocket in $K_{neu}$_free_1 and $Mg_{0.3}$_free_1. $d_{U20-A23}$ is shown in yellow dashed line, and two key metal ions are also shown in color spheres (K$^+$ in orange and Mg$^{2+}$ in green). **b** Detailed structure of $C_{J12-P1}$ in $Mg_{0.3}$_free_1. Mg$^{2+}$ interacts with the surrounding residues through its chelated water molecules. **c** Arrangement of hydrated Mg$^{2+}$ ions in the groove formed by stems P2 and P3 (the two strands connected by J23). **d** The heat maps generated by computing the number of common metal ions in the range of 6 Å around every two phosphate groups for all eight simulation trajectories. The color bar shows the variations in number of metal ions from 0 (black) to 4 (white). **e** The electrostatic potential surface of AARA structures from experiment and $K_{neu}$_free_1. The coloring range of electrostatic potential is from −40 $k_BT·e^{-1}$ (red) to 0 (blue). **f** Detailed view of an important K$^+$ ion binding site in $K_{neu}$_free_1. This site is located in the middle of J23 and J12, and is formed by residues U20, A21, U48, and U49. K$^+$ ion in this site will interacts with five oxygen atoms (represented by red spheres), namely O2 and O2′ on U20, OP1 on A21, O4 on U48, as well as OP1 on U49, respectively.

Snapshot indicates that even if one $Mg^{2+}$ ion enters this site, it still keeps hydrated state and forms stable hydrogen bond interaction with AARA structure through these six chelated water molecules, which is the so-called outer-sphere coordination effect (Fig. 4b)[41,53]. In details of this binding event, $Mg^{2+}$ ion mainly bridges the electronegative atoms on the backbone of this region, including OP2 atoms from C18, A19, U74, and OP1 from A73, to make the structure of this region more compact, i.e., the distance between U20 and A23 is only about 8.5 Å. Notably, the flexibility of structure around this site would not disappear completely due to the binding of $Mg^{2+}$ ion. On the contrary, the $Mg^{2+}$ ion bound here will fluctuate with the structure of this region (Supplementary Movie S1). In addition, since the strength of outer-sphere coordination is weaker than that of inner-sphere coordination, a rapid exchange phenomenon of $Mg^{2+}$ ions is observed to occur at this site (Supplementary Movie S2). Even so, the above phenomena does not essentially affect the structure here.

As mentioned in previous mix-and-inject XFEL experiment[35], in the absence of ligand, concerted movement of the hinge (U22, A23) and latch (U48, U49, U51) regions results in considerable narrowing of the major groove formed by helices P1 and P3, which is about $9.3 \pm 0.3$ Å for ligand-free structure (measures ~16.6 Å for ligand-bound structure). As discussed above, because the strong electrostatic repulsion between J31 and P1 can be effectively weakened by $Mg^{2+}$ ions, the width of the groove formed by them also decrease obviously. Here, to make a better comparison with the experiment, the width of this major groove was measured as the distance between residue A19 and residue U71. In the presence of $Mg^{2+}$ ions, the width of this groove is greatly compressed to about 14 Å, which is smaller than any other condition containing only $K^+$ ions (both ligand-free and ligand-bound states) (Supplementary Table S1). Beyond that, the narrowing of the groove between P2 and P3 (extending to J23) is also caused by the regular arrangement of hydrated $Mg^{2+}$ ions in it (Fig. 4c).

$K^+$ ions addition alone do not only cause the loosening of the $C_{J12-P1}$ and J31 structure, but also further lead to the proximity between P1 and J23. This event will create a low electrostatic potential hole between P1 and J23, which is very suitable for a single $K^+$ ion residence (Fig. 4e). In detail, five negatively charged atoms in this region interact directly with $K^+$ ion (the distance between each of them and $K^+$ ion is less than 3 Å), they are O2 and O2' on U20, O4' on A21, OP1 on U48, as well as O4 on U49, respectively (Fig. 4f). Conversely, the binding of $K^+$ ion in this region also promotes the stability of this structural state. Additionally, we have observed the rapid ion exchange event at this $K^+$ binding site, and this does not affect the stability of this region (Supplementary Movie S3).

**Effect of monovalent ionic strength on the structure of AARA.** As depicted in the results of free energy landscape for AARA structure at different ionic conditions, the increase of $K^+$ concentration may contribute to the formation of compact $C_{J12-P1}$ structure to a certain extent. Fortunately, we have also found the allosteric phenomenon of $C_{J12-P1}$ in one of $K_{0.3}$_free systems (Fig. 5a). In $K_{0.3}$_free_2, $d_{U20-A23}$ fluctuated around a value of about 12 Å before 740 ns, then, it dropped sharply to about 8.5 Å and maintained until the end of the simulation. Clearly, $\varphi C$ of phosphate group in the trajectories before and after 740 ns of $K_{0.3}$_free_2 are obviously different in the regions P1 (5' end) and J31, suggesting the change of the compactness of $C_{J12-P1}$ has a direct effect on the electrostatic potential of RNA structure (Fig. 5b). The lower $\varphi C$ of phosphate group, the more cations can be attracted to stay here. However, the increase of compactness also aggravates the electrostatic repulsion between phosphate

groups in these regions. Therefore, it needs enough cations to gather here to maintain the stability of the compact structure.

In $K_{neu}$_free system, neither loose nor compact (obtained from the trajectory with $d_{U20-A23}$ constrained at 8.5 Å in umbrella sampling simulation) $C_{J12-P1}$ structure can attract enough $K^+$ ions to counteract this repulsive effect (Fig. 5b). In $K_{0.3}$_free_2, in addition to counterions, excess $K^+$ ions in the salt solution participate in the positive compensation for the high negative charge density of $C_{J12-P1}$. For a more intuitive understanding, we calculated the total cumulative electrostatic potential strength coefficients for $K_{neu}$_free_1 and $K_{0.3}$_free_2 (Fig. 5c), and colored them in three typical corresponding structures (Fig. 5d). The total $\varphi C$ represents the superposition electrostatic potential effect of all important charged elements at a certain position (excluding the phosphate group itself here) on the backbone of RNA. An electronegative phosphate group must be unstable in a negative potential environment, hence the stability of phosphate backbone of RNA can be judged in terms of the total $\varphi C$. In $K_{neu}$_free system, the whole backbone of AARA is immersed in the negative potential environment (average value of total $\varphi C$ is about $-0.239 \, C \cdot Å^{-1}$), which results in a loose backbone of AARA structure, especially in the regions of $C_{J12-P1}$ and ligand binding pocket. In contrast, even though the effect of $Cl^-$ ion is considered, each phosphate group is in a nearly neutral environment for $K_{0.3}$_free_2 before and after 740 ns (average values of total $\varphi C$ are about $-0.007 \, C \cdot Å^{-1}$ for before 740 ns and $-0.004 \, C \cdot Å^{-1}$ for after 740 ns). As a result, two kinds of low-energy structures can exist stably in the $K_{0.3}$_free system. Moreover, the time evolutions of $\varphi C$ for $K^+$ ions (positively charged) and phosphate groups (negatively charged) at each residue position in $K_{0.3}$_free_2 also verify the strong correlation between the density of metal ion and the transition of AARA structure (accompanied with the change of electrostatic potential in different regions) (Supplementary Fig. S6). It is worth mentioning that once compact $C_{J12-P1}$ structure appears, the low electrostatic potential hole between P1 and J23 collapses, and the $K^+$ ion in it will regain freedom (Supplementary Movie S4).

**Ligand binding affinities in $Mg^{2+}$-free environment.** As a riboswitch, ligand binding will inevitably change the structure of AARA. To eliminate the effects of $Mg^{2+}$ ion, we investigated the interaction between ligand and AARA in the environment which is only neutralized by monovalent metal ions. Generally speaking, similar to the $K_{0.3}$_free system, the existence of ligand also improved the stability of a compact $C_{J12-P1}$ structure to a certain extent. However, the PMF profiles of $K_{neu}$_bound system indicated that the compact $C_{J12-P1}$ structure is still likely to return to the loose state, which may be attributed to the repulsion of high negative charge density in the compact state.

The two independent trajectories of $K_{neu}$_bound system show totally different structural fluctuation features. In $K_{neu}$_bound_1, $d_{U20-A23}$ is maintained at around 12 Å, suggesting the existence of a loose $C_{J12-P1}$ structure. In contrast, the structural transition of $d_{U20-A23}$ between 8.5 Å and 12 Å appears repeatedly in $K_{neu}$_bound_2, and the compact $C_{J12-P1}$ is the main one. By examining these two trajectories in detail, we found that the structure of ligand binding pocket (with compact $C_{J12-P1}$) in $K_{neu}$_bound_2 was very similar to that in the environment with $Mg^{2+}$ ions. Because the ligand can link the bases of four uracil residues (U22, U47, U51, and U74) together and form an aromatic ring plane, the adjacent base pair A21-U75 has a greater chance to form a stable spatial structure with the binding pocket through π-π stacking interaction. Unfortunately, due to the strong electrostatic repulsion caused by the increase of compactness, this configuration cannot be sustained in low-concentration $K^+$ ion environment. Without the aid of metal

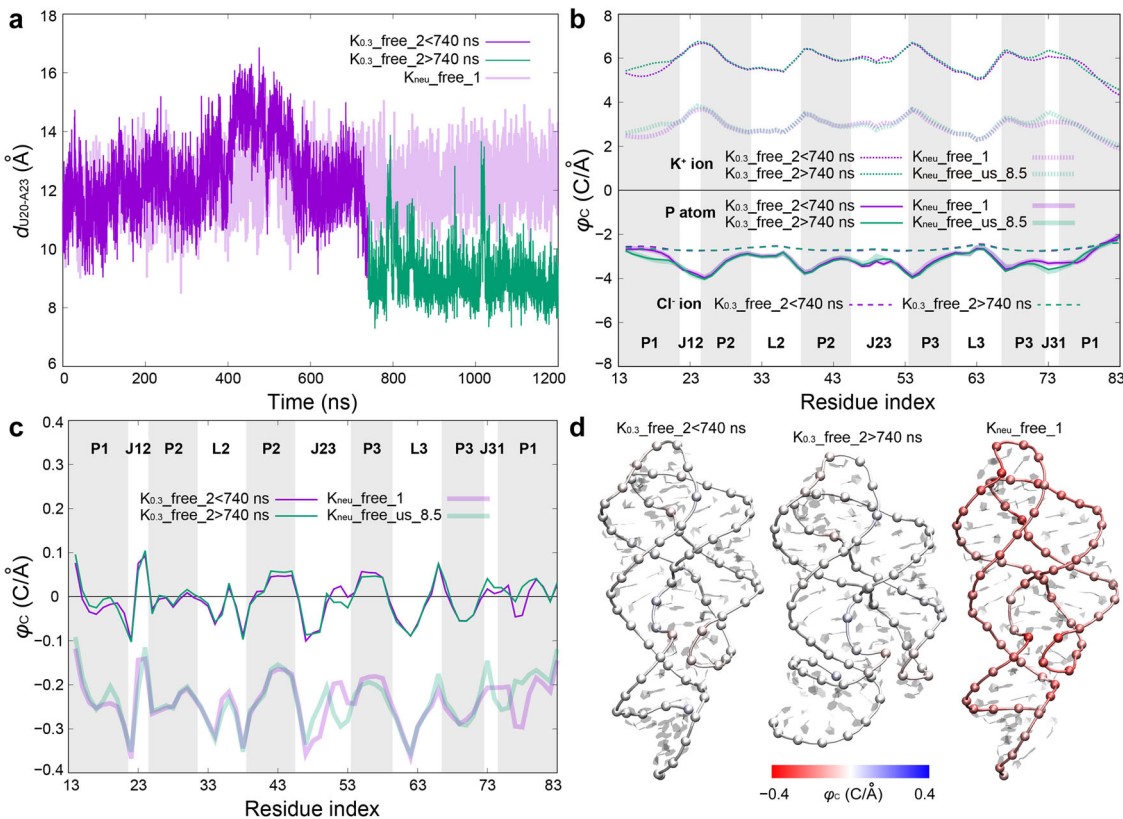

**Fig. 5 Effect of monovalent ionic strength on the structure of AARA. a** Time evolutions of the distance between U20 and A23 for $K_{neu\_free\_1}$ (light purple) and $K_{0.3\_free\_2}$ (<740 ns in purple and >740 ns in green). **b** The cumulative electrostatic potential strength coefficients $\varphi C$ of phosphate groups (solid lines), $K^+$ ions (dense dashed lines) and $Cl^-$ ions (sparse dashed lines) in different regions of AARA structure. According to the $d_{U20\text{-}A23}$, the calculation of $\varphi C$ for $K_{0.3\_free\_2}$ was divided into two parts: simulation time <740 ns and >740 ns (represented by thin lines), and the results of them were compared with those of $K_{neu\_free}$ system (taking $K_{neu\_free\_1}$ as an example, in purple thick lines) and $K_{neu\_free\_us\_8.5}$ (the case where $d_{U20\text{-}A23}$ is constrained at 8.5 Å, in green thick lines), respectively. **c** The total cumulative electrostatic potential strength coefficients $\varphi_C$ in different regions of AARA structure. **d** The total cumulative electrostatic potential strength coefficients $\varphi_C$ are shown in three typical corresponding structures, and the color bar shows the variations in $\varphi_C$ from $-0.4$ C·Å$^{-1}$ (red) to $0.4$ C·Å$^{-1}$ (blue).

ions, it is difficult for $C_{J12\text{-}P1}$ to keep compact even if there is an offset face-to-face π-π stacking effect between A21-U75 base pair and ligand binding plane. Once $C_{J12\text{-}P1}$ returns to the loose structure state, the stacking structure around the binding pocket collapses. As a result, the hydrogen bonds of A21-U75 base pair become unstable and even break, and the free U75 tends to tilt up and is no longer parallel to the ligand binding plane (Fig. 6a).

Clearly, this situation will affect the binding stability between the ligand and the pocket. To quantitatively estimate this effect, we calculated the binding free energy between ligand and AARA using MM-PBSA method. The absolute binding free energies at 300 K in $K_{neu\_bound\_1}$ and $K_{neu\_bound\_2}$ are totally different ($\Delta G$ ~1.3 kcal·mol$^{-1}$ and ~$-4.8$ kcal·mol$^{-1}$), and our calculations indicate that the entropic contributions for these two cases are very close and unfavorable ($-T\Delta S$ ~ $-18.1$ kcal·mol$^{-1}$ and ~$-18.2$ kcal·mol$^{-1}$). As a consequence, we conclude that the deformation of $C_{J12\text{-}P1}$ does affect the ligand binding, and it can be attributed to the enthalpy contributions ($\Delta H$ ~ $-16.8$ kcal·mol$^{-1}$ and ~$-23.0$ kcal·mol$^{-1}$). To quantitatively confirm the difference of binding mode between $K_{neu\_bound\_1}$ and $K_{neu\_bound\_2}$, we decomposed the enthalpy contribution into two energetic terms: electrostatic energies $\Delta H_{ele+pol}$ and non-electrostatic energies $\Delta H_{vdW+nonpol}$. The calculations indicate that the electrostatic energies in $K_{neu\_bound\_1}$ (~9.2 kcal·mol$^{-1}$) is more unfavorable than that in $K_{neu\_bound\_2}$ (~4.1 kcal·mol$^{-1}$), and the non-electrostatic

energies in $K_{neu\_bound\_1}$ (~$-26.0$ kcal·mol$^{-1}$) is marginally less favorable than that in $K_{neu\_bound\_2}$ (~$-27.1$ kcal·mol$^{-1}$). Further, we examined the contribution of each residue to these two energetic terms to explore the effect of the detailed structural changes around the binding pocket on the ligand binding stability (Fig. 6b, c). The results showed that residues A21, U22, U47, and U75 have the most influence on ligand binding during the two-state transition of $C_{J12\text{-}P1}$, and their contributions to the binding free energy are different. As discussed in previous section, a loose $C_{J12\text{-}P1}$ leads to the proximity of J12 and J23, resulting in an increase of electronegative density in the nearby region. Thereby, the electrostatic contribution of the above-related residues (A21, U22, and U47) to ligand binding is obviously more unfavorable in $K_{neu\_bound\_1}$. Equally notable is that the destruction of A21-U75 causes U75 to lose the non-electrostatic interaction (mainly through π-π stacking, see Supplementary Method S1 for evaluation criteria) with U74, while A21 still maintains its π-π stacking interaction with A76 and is not affected so much (Fig. 6a and Supplementary Table S2).

**The impact of structural changes around ligand binding pocket on the EP**. To evaluate the effect of metal ions on the structure of ligand binding pocket, we calculated the $R_G$ of four important residues (U22, U47, U51, and U74) (Fig. 7a). The results demonstrate that the structure of binding pocket has following

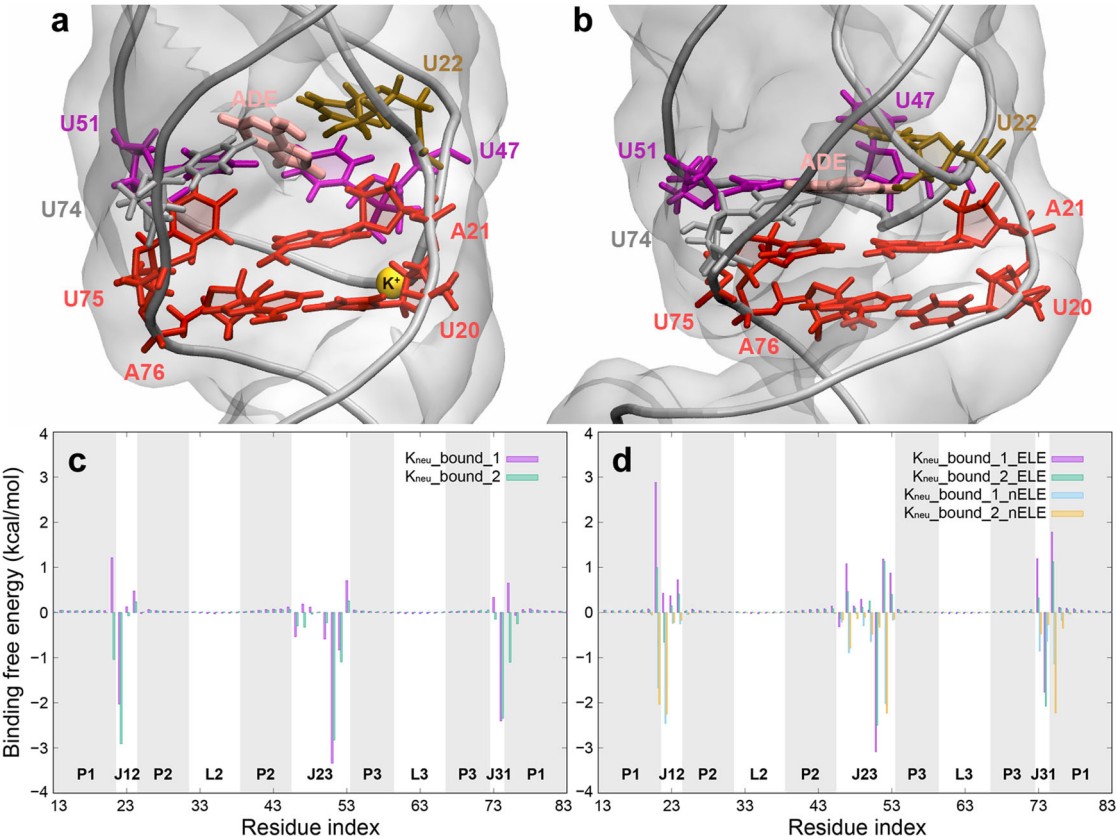

**Fig. 6 The binding pocket structure and ligand binding affinities in Mg²⁺-free environment. a** The snapshots of typical structures near ligand binding pocket in $K_{neu}$_bound_1. Because $C_{J12-P1}$ is almost always in a loose state in $K_{neu}$_bound_1, we have marked the important $K^+$ ion binding site (between P1 and J23, represented by orange sphere) in this state. **b** The snapshots of typical structures around ligand binding pocket in $K_{neu}$_bound_2. **c** The contribution of each residue to the free energy (ligand binding to the AARA) in $K_{neu}$_bound_1 and $K_{neu}$_bound_2. **d** The contribution of each residue to the free energy is decomposed into electrostatic and non-electrostatic terms in $K_{neu}$_bound_1 and $K_{neu}$_bound_2.

characteristics under different ionic conditions: (i) when the ligand is not bound, the fluctuation of binding pocket structure in $K^+$ ion environment is extremely large, and it has little relationship with the change of concentration of $K^+$ ion; (ii) ligand binding greatly stabilizes the pocket, and it will expand slightly to accommodate the ligand; (iii) as mentioned previously, the looseness of $C_{J12-P1}$ leads to the proximity of J12 and J23, which also reduces the $R_G$ of binding pocket; (iv) even without ligand binding, $Mg^{2+}$ ions can also restrain the structure of binding pocket, and even make it more compact than in other ionic environments. As for the fourth feature, how is the compact pocket structure formed in the $Mg^{2+}$ ion environment? In detail, U22 and A52 form a stable Watson-Crick base pairing, and it makes U51 constrained and unable to swing out of the binding pocket easily. Meanwhile, even without ligand binding, A21-U75 base pair is quite stable in $Mg^{2+}$ ion environment, and U74, which forms a π-π stacking with U75, tends to stay in the binding pocket. Fortunately, without the barrier of ligand, the above constraints give U51 and U74 the opportunity to approach, and finally, relatively stable hydrogen bond interactions are formed between them (Fig. 7b and Supplementary Fig. S7).

A stable P1 structure can release the Shine–Dalgarno and initiation codon sequences to facilitate the translational process, and once P1 collapses, these segments are sequestered through base pairing interactions. Thus, the key to understand the regulation mechanism of *add* adenine riboswitch is to clarify the structural correlation between binding pocket and P1 stem. Here, we still use the states of $C_{J12-P1}$ to characterize the structural

features of binding pocket, and investigate the influences of the two states of $C_{J12-P1}$ on the structure and stability of P1. Based on previous analysis, the deformation of $C_{J12-P1}$ is caused by the interaction between the negatively charged phosphate groups on its backbone and the metal ions. Our results show that the distance between each of the two adjacent residues on $C_{J12-P1}$ varies from one to another, with little overall difference under different ionic conditions, suggesting that the adjacent residues in this region do not become close to each other due to the binding of $Mg^{2+}$ ion (Supplementary Fig. S8a). Thus, the deformation of $C_{J12-P1}$ is more likely to come from the bending of backbone. According to the results of the bending angle of backbone, the addition of $Mg^{2+}$ ion does make the backbone at $C_{J12-P1}$ obtain greater bending than that in pure $K^+$ ion environment. Furthermore, the main difference of $C_{J12-P1}$ structure under different ionic conditions comes from the curvature formed by residues A21, U22 and A23. In $Mg_{0.3}$_free system, the angle of this curvature can be maintained stably at about 60 degrees, which is much smaller than that in $K_{neu}$_free system (about 100 degrees) (Supplementary Fig. S8b). The sharply bent structure of $C_{J12-P1}$ makes the bases in this region compact together (Fig. 7c), which can protect them from the invasion of water molecules (Characterized by SASA) and keep stable π-π stacking interactions (especially for the bases of residues A21 and U75, see Supplementary Table S3). In addition, in the absence of $Mg^{2+}$ ions, PMF results indicate that both the increase of $K^+$ ion concentration and the aromatic ring plane formed by ligand binding can delay the disorganization of the stacking structure in this region.

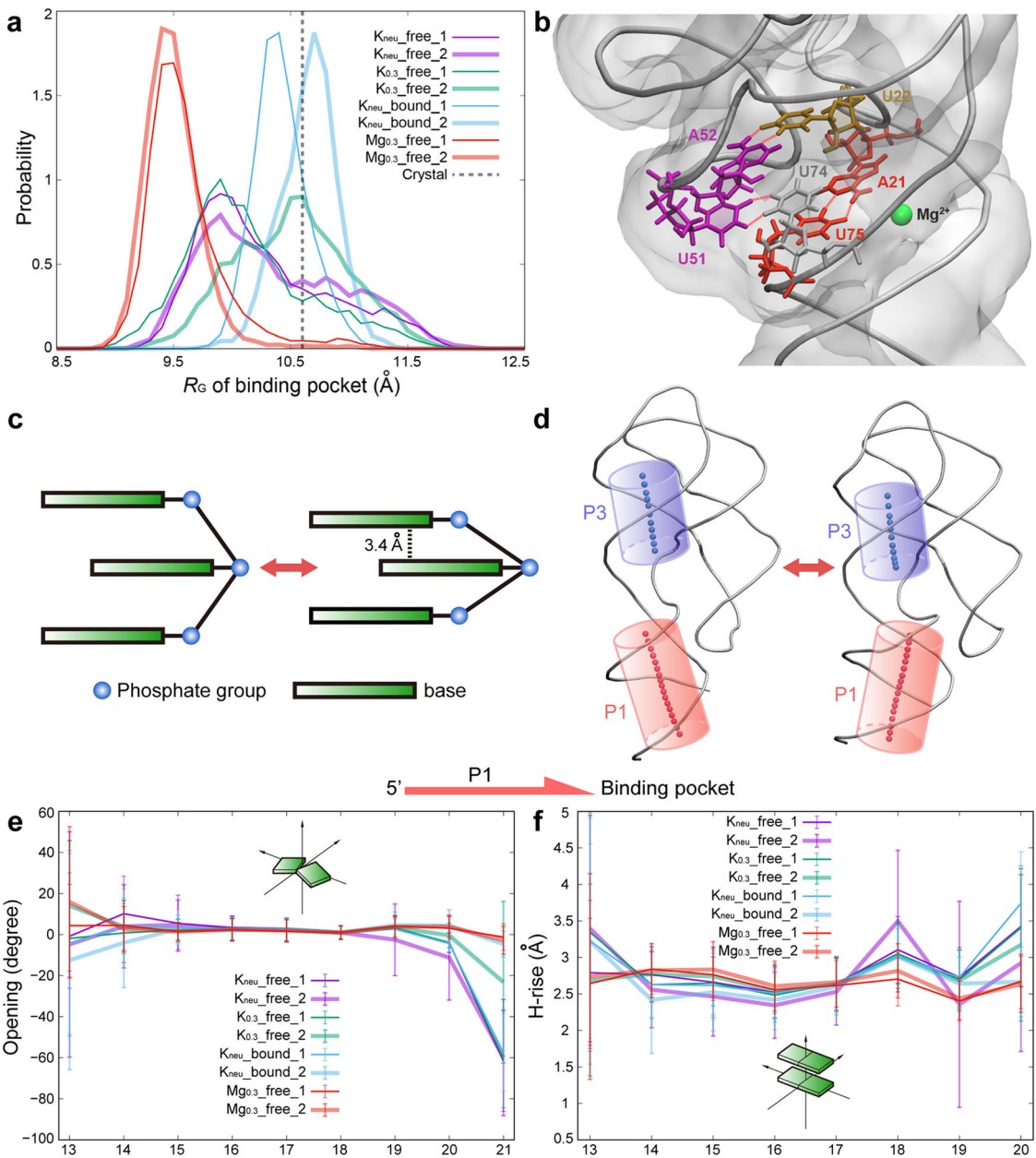

**Fig. 7 The structural changes around ligand binding pocket and their impacts on other regions of AARA. a** The normalized distributions of $R_G$ of binding pocket for all eight simulation trajectories, and the gray dashed line corresponds to $R_G$ of binding pocket in the crystal structure (~10.6 Å). **b** The structure of ligand binding pocket in $Mg_{0.3}$_free system (taking $Mg_{0.3}$_free_1 as an example), and the hydrogen bond interactions between important residues near binding pocket are indicated by red dashed lines. An important $Mg^{2+}$ ion binding site (represented by green sphere) in this state is also marked in the figure. **c** A schematic diagram of change in base stacking structure caused by the bending deformation of the backbone at $C_{J12-P1}$. **d** The change of coaxial stacking morphology between P1 and P3 caused by the deformation of $C_{J12-P1}$. The central axes of P1 (red) and P3 (blue) are drawn with the van der Waals representation of hydrogen atom. The average values and standard deviations of Opening (**e**) and H-rise (**f**) of each base pair in P1 helix for all eight simulation trajectories, and the direction of P1 (from 5' end to binding pocket) is also shown in the figure.

In any case, the looseness of $C_{J12-P1}$ will further affect the structure and stability of P1. For simplicity, we use two typical parameters Opening and H-rise to describe the structural changes of P1 helix (Fig. 7e, f), and both of them can be obtained from the analysis results of Curves+ program (the data of P3 are extracted as a comparison, see Supplementary Fig. S9). Opening is used to evaluate the pairing health within each base pair, and H-rise is used to measure the relative motion between consecutive base pairs. Firstly, the elongation of $C_{J12-P1}$ will greatly increase the probability of base pair A21-U75 being destroyed (Opening is negative), and this

destruction may spread to the interior of P1 stem. Secondly, the fluctuation of the relative position between several base pairs close to the binding pocket increases, and this part of the P1 helix is also stretched, which may lead to the weakening of the stacking effect between adjacent base pairs. Simultaneously, the coaxial stacking morphology between P1 and P3 is also closely related to the structure of $C_{J12-P1}$ (Fig. 7d and Supplementary Table S4). Once $C_{J12-P1}$ elongates, P1 and P3 will move away from each other and their central axes will shift, resulting in a decrease of coaxial stacking stability between them.

**Structural dynamics of AARA at different ionic conditions.**
Generally, an RNA molecule relies on a structural correlation network mainly composed of hydrogen bond interactions to maintain its functional structure and morphology, in which metal ions play an auxiliary role. If there are stable hydrogen bond interactions between two residues, the motion between them should be synergistic. At the same time, the metal ions dispersed in the RNA structure will also affect the structural correlation network. Thus, mapping the structural correlation network is very helpful to understand the dynamic properties of AARA. To probe the feature of the structural correlation network in detail, we calculated the correlation coefficient of both direction and magnitude of velocity vector between any two residues (Supplementary Method S2). Because the real-time conformations (the molecular structure at last moment) are used as the reference state to generate the velocity vector, our analyses have higher accuracy than the traditional dynamic cross-correlation matrices (DCCM) method (just taking average structure as reference state).

The correlation maps of direction and magnitude of velocity vector (Fig. 8a and Supplementary Fig. S10) both indicate that $Mg^{2+}$ ions greatly improve the correlation between residues in two regions: (i) the groove between J31 and one strand (5' end) of P1; (ii) the groove formed by stems P2 and P3 (the two strands connected by J23). As described earlier, these two regions have low electrostatic potential and appropriate space matching with hydrated $Mg^{2+}$ ions, so they are very suitable for long-term residence of $Mg^{2+}$ ions. Simultaneously, $Mg^{2+}$ ions link the residues on both sides of these grooves together by bridging interactions. For the other three systems without $Mg^{2+}$ ions, their correlation maps are very similar, showing strong motion correlations in the regions with dense hydrogen bond interaction (especially for the stems P1, P2 and P3). Interestingly, $Mg^{2+}$ ions may not only promote the correlation strength between some residues, but also can weaken the dynamic synergy in above regions. In $Mg_{0.3}$_free system, the structural correlations between residues in all three helical stems decreased. In fact, it is those $Mg^{2+}$ ions bound in the groove (between J31 and P1, as well as between P2 and P3) that establish new links for nearby residues and weaken the movement synergy between residues within the stems (obviously for P2 and P3), which nevertheless makes the AARA more structural integrity. Moreover, regardless of the ionic environment, residue A24 has a strong dynamic correlation with base pair C54-G72 in P3 (near the binding pocket). Previous studies have shown that A24 is located downstream of P2 and sandwiched between residues A73 and C54-G72, acting like a hook holding stems P2 and P3 together[34]. The stable correlation of A24 with C54-G72 indicates that the near-pocket ends of P2 and P3 are always locked together with high structural stability, which implies that the ionic environment has no marked effect on the upstream structure of binding pocket.

We have also calculated the motion fluctuation of each residue to explore the stability of different regions of AARA (Fig. 8b). The results of RMSF indicate that, overall, AARA shows stronger stability in $Mg^{2+}$ ion environment than in the other cases. For different regions, the change of ionic condition has little effect on L2, L3, P2 and P3, but had some great impacts on the stability of P1 and the junction region (J12, J23 and J31). In fact, the direct connection of P1 to the junction region causes its structural fluctuation to be closely related to $C_{J12-P1}$ (especially in

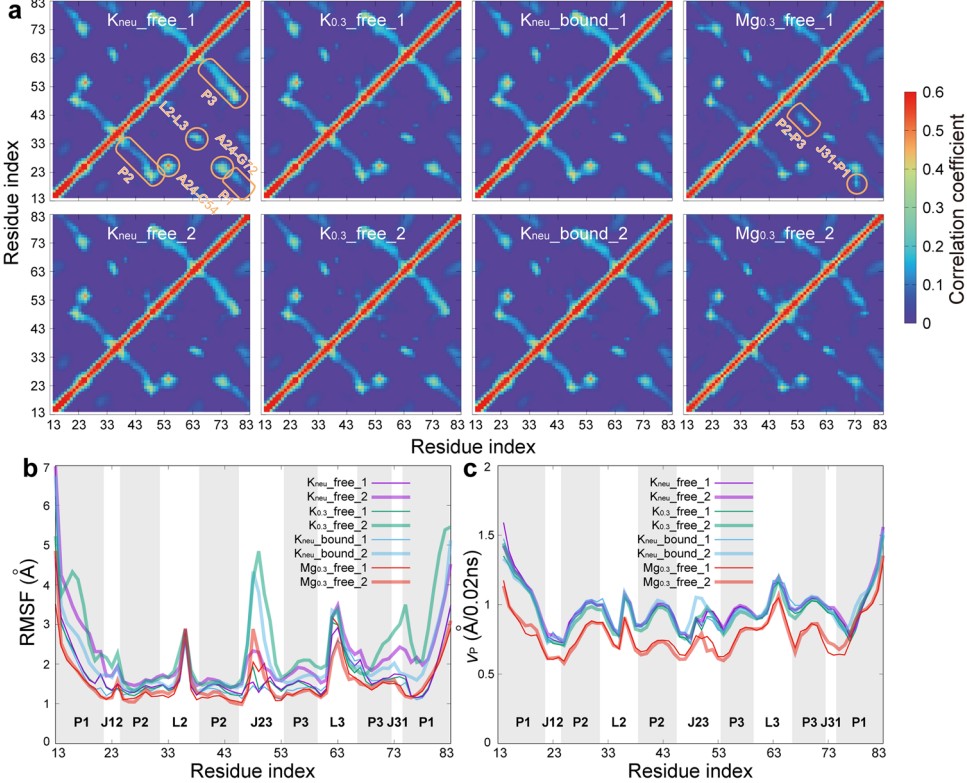

**Fig. 8 Structural dynamics of AARA. a** The correlation maps of direction of velocity vector between any two phosphate groups for all eight simulation trajectories. The color bar shows the variations in correlation coefficient from 0 (dark blue) to 0.6 (red). Several regions showing strong correlation are marked with orange boxes. **b** The RMSFs of AARA structure for all eight simulation trajectories. The sampling interval is 20 ps; thus, a total of 60,000 conformations are used to calculate the RMSFs for each trajectory. **c** The velocity of phosphate group on each residue for all eight simulation trajectories. Since the sampling interval is 20 ps, we simply use the displacement of phosphate group in this interval to replace its distance in this interval. Then, the displacement divided by time interval can be used to obtain the approximate velocity of each residue.

$K_{0.3}$\_free\_2 and $K_{neu}$\_bound\_2). Unexpectedly, although both ends of J23 are connected to structurally stable P2 and P3, it still shows abnormal instability in $K_{0.3}$\_free\_2 and $K_{neu}$\_bound\_2. Because $C_{J12-P1}$ has also undergone great structural changes in these two trajectories, it is reasonable to suspect that the cause of the above anomalies is related to this factor. We have compared the evolution of RMSD of J23 with that of $d_{U20-A23}$ over time, and found that there was a strong coupling between them. By checking these two trajectories (particularly for the dominant motion pattern in $K_{0.3}$\_free\_2, see Supplementary Fig. S11 and Movie S5), we confirm that there is an obvious volume repulsion effect between $C_{J12-P1}$ and J23, which leads to the change of J23 structure with the deformation of $C_{J12-P1}$ (Supplementary Fig. S12). In addition to these, the abnormally high values at P3-J31 (as well as J12) in RMSF results imply that the stacking between base pair A24 on J12 and C54-G72 on P3 may not be fixed (appearing in $K_{0.3}$\_free\_2), and that A24 (involving A73 stacked below it) switches the stacking pattern between the two bases (C54 and G72), causing the backbone here to be distorted as well (Supplementary Fig. S13 and Movie S6).

Moreover, we have calculated the velocity of phosphate group on each residue to further investigate the dynamical features of different regions of AARA (Fig. 8c). Results clearly show that $Mg^{2+}$ ions greatly slow down the kinetics of AARA fluctuations, especially in P2, P3 and some regions close to $C_{J12-P1}$ (J12, J31 and the strand near the 5' end of P1). As mentioned previously, there is a similarity between these regions, that is, they all have grooves that can stably bind $Mg^{2+}$ ions. Under the bridging interactions of $Mg^{2+}$ ions, these regions can no longer move independently and freely, which may be the reason why $Mg^{2+}$ ions slow down the movement of them.

## Discussion
Metal ions play an important role in the process of gene regulation by riboswitches. In fact, the cellular environment in which mRNA is located is complicated, and cations as well as metabolic ligands are all free in the cytoplasm around riboswitch fragments. Therefore, it is meaningful to give a comprehensive and detailed understanding of the mechanisms by which the above various factors affect the structure and dynamics of riboswitches. Here, taking AARA as an example, we have explored the nature of metal ions on the structure and function of these special RNA elements at atomic level using MD simulation. The present results indicate that metal ions (particularly for $Mg^{2+}$) can effectively change the local structure of AARA (especially at the connection between J12 and P1) through specific binding, and potentially affect the execution of relevant functions. Besides, the change of $K^+$ concentration and ligand binding can also influence the structure of the P1 stem directly connected to the EP. In general, the knowledge gained from the present studies can be summarized by a simple dynamic model (Fig. 9). Firstly, $C_{J12-P1}$ adopts a loose state under the condition of low $K^+$ concentration. Secondly, with the increase of $K^+$ concentration or the addition of cognate ligand, $C_{J12-P1}$ has the opportunity to form a compact structure, but it cannot be locked in this state only by electrostatic neutralization of high concentration of $K^+$ or the interaction between ligand and binding pocket. Finally, the binding of $Mg^{2+}$ at $C_{J12-P1}$ tightly constrains the structure of this region, and also promotes the perfect binding of ligand in the pocket. In addition, the compact $C_{J12-P1}$ can improve the stability of P1 from the following two aspects: (i) the compressed space in this region forces A21-U75 to stack with those bases in the binding pocket, so that this base pairs are not easy to be destroyed; (ii) The bent $C_{J12-P1}$ can hold P1, making it form a more stable coaxial stacking interaction with P3.

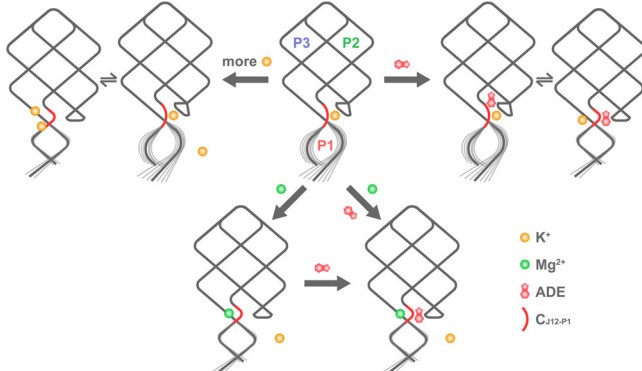

**Fig. 9 Simple dynamic model for the structural transition of AARA (with or without ligand binding) in different ionic environment.** Firstly, $C_{J12-P1}$ adopts a loose state under the condition of low $K^+$ concentration. Secondly, increasing the concentration of $K^+$ or ligand binding both has the opportunity to make $C_{J12-P1}$ enter the compact state. Finally, the addition of $Mg^{2+}$ ions can stably maintain the compact $C_{J12-P1}$ structure. Additionally, there is a strong structural correlation between $C_{J12-P1}$ and P1. Once the structure of $C_{J12-P1}$ collapses, the stability of P1 will be greatly reduced.

Actually, only in the environment containing $Mg^{2+}$, the structure of the *add* adenine riboswitch has small fluctuation and is easy to be captured. Thereby, $Mg^{2+}$ ions have been added to almost all relevant experiments to help determine the structure of this riboswitch[28,29,35]. For this reason, special attention should be paid to controlling the metal ion concentration (especially for $Mg^{2+}$) in MD simulations for a reasonable comparison with the experiments. In our simulations containing $Mg^{2+}$, although we added $Mg^{2+}$ ions to the solvated box according to the concentration of 300 mM, due to the high density negative charge on the RNA backbone, $Mg^{2+}$ ions have to gather around riboswitch quickly, resulting in its bulk concentration lower than the preset concentration. The estimated results showed that the bulk concentrations of $Mg^{2+}$ in $Mg_{0.3}$\_free\_1 and $Mg_{0.3}$\_free\_2 are about 225 mM and 226 mM, respectively (Supplementary Method S3 and Table S5), which is little different from the value (200 mM $Mg^{2+}$) in several existing high-precision X-ray diffraction experiments[28,29]. Generally, our present results are in accordance with the previous experimental findings, indicating high reliability of the corresponding conclusions. Firstly, the SAXS profiles and corresponding Kratky plots calculated based on our MD simulation trajectories are highly similar to the results obtained by XFEL serial crystallography[35], especially the Kratky plots of AARA in all eight simulation trajectories show asymmetric Bell-shape curves, characteristic of a folded structure with partial flexibility (Supplementary Method S4 and Fig. S14). Secondly, the strong binding sites of $Mg^{2+}$ predicted from our MD simulations are quite consistent with the positions of $Mg^{2+}$ ions captured in several X-ray diffraction experiments (Supplementary Fig. S15). Notably, almost all $Mg^{2+}$ ions closely bound to the riboswitch maintain their first hydration shell during the course of simulations we conducted. This phenomenon is somewhat different from that observed in the crystal structures obtained by relevant experiments (in particular, $Mg^{2+}$ ions form inner-sphere contacts with OP2 on A23 and OP1 on A24), nevertheless, it does not essentially affect our conclusions since the $Mg^{2+}$ ions in the simulations are stably bound to similar positions through outer-sphere contacts. The reason for this is that the $Mg^{2+}$ ion model we used underestimates the water exchange rate of its first hydrated layer[54], resulting in their ability to rapidly approach the binding sites but not effectively form inner-sphere contacts with the phosphate groups in a short time. In this regard, some new

force fields[54,55] have been improved for this and can fit the experiments better, and although their robustness has not been widely verified, it is still worth trying and comparing with existing results in the subsequent studies. In fact, a recent rigorous MD work showed a good agreement between the results of several currently popular force field combinations of RNA and $Mg^{2+}$ ion for small riboswitches[56]. Thirdly, the binding free energy of ligand to AARA (~ −4.8 kcal·mol$^{-1}$ in $K_{neu}$_bound_2, $C_{J12-P1}$ remains compact most of the time) under low $K^+$ ion concentration environment is in the same order of magnitude as the results from our previous MD simulations (~−8.2 kcal·mol$^{-1}$)[46] and other experimental measurements (~−8.8 kcal·mol$^{-1}$)[57,58]. In general, accurate estimation of entropy in MD simulation is a difficult task. For the sake of rigor, we have estimated conformational entropy using quasi-harmonic models in cartesian coordinates[59], and obtained the entropy of $K_{neu}$_bound_1 and $K_{neu}$_bound_2 are −25.1 kcal·mol$^{-1}$ and −25.4 kcal·mol$^{-1}$, respectively, showing a good agreement with the experimental results[57]. Nevertheless, since the enthalpy contribution was also underestimated in MM-PBSA calculation, the free energy obtained based on normal mode approximation did not deviate much from the experiments. Given that the contribution of entropy is almost identical for both $K_{neu}$_bound systems regardless of the method used, our results therefore clearly confirm that the ligand binds most tightly to the AARA only in the presence of $Mg^{2+}$ ions (attributed to a more compact stacking pattern at the binding pocket).

Another valuable thing is that we have given the possible conformation of AARA in the absence of $Mg^{2+}$ ion. This kind of conformation is not quite stable, which may lead to the collapse of P1 stem and put the *add* adenine riboswitch into the translation termination state. Moreover, a strong $K^+$ binding site between P1 and J23 is found in this conformation, and the $K^+$ ion bound here may prevent the structure of AARA from returning to the compact state. Although the increase of $K^+$ concentration will make it possible for the compact AARA structure to appear, as a highly mobile monovalent ion, $K^+$ mainly form an ion atmosphere (rather than specific binding) to neutralize the negative charge on the backbone of nucleic acids (gives an average value of total $\varphi C$ of about 0 in $K_{0.3}$_free system, see Fig. 5c). In contrast, the average value of total $\varphi C$ in $Mg_{0.3}$_free system is about +0.143 C·Å$^{-1}$ (taking $Mg_{0.3}$_free_1 as an example, see Supplementary Fig. S16), which suggests that $Mg^{2+}$ ions have a stronger ability in negative charge screening compared to $K^+$ ions. Together, all these further confirm that the participation of $Mg^{2+}$ ion is a necessary factor to promote the initiation of translation. In fact, the roles of $Mg^{2+}$ and cognate ligand are complementary. The three-way junction forms a more compact structure under the bridging interactions of $Mg^{2+}$ ions to restrict the ligand in this region, and because of this, the ligand can be stably bound here and serve as a key factor to keep this riboswitch in the translation activation state.

Limited by computing power, the duration of MD simulation is finite. In the whole process of our simulation, due to the abundance of hydrogen bond interactions, the responses of hairpin loops (L2 and L3) and related stems (P2 and P3) to the change of ionic conditions are not obvious. However, this result reflects that the structural sensitivity of ligand binding pocket to metal ions is much higher than that in other regions of *add* adenine riboswitch. Besides, since the binding pocket of AARA spontaneously relaxes to the loose state in pure $K^+$ ionic environment, conversely, can $Mg^{2+}$ ions restore it to the compact state again? To answer this question, an extra MD simulation was performed using a typical conformation extracted from the $K_{neu}$_free system as initial structure and protocol as the previous $Mg_{0.3}$_free system, and the production time remained 1.2 μs. The results show that $d_{U20-A23}$

decrease to ~8.5 Å after about 400 ns, and since then, the evolution characteristics of RMSD are almost the same as those in $Mg_{0.3}$_free system (Supplementary Fig. S17a, b), suggesting the recovery of compact binding pocket and stable P1 stem. Moreover, the strong correlation between $Mg^{2+}$ and structure of AARA in this simulation means that $Mg^{2+}$ ions play a key role in the structural transition of riboswitch (Supplementary Fig. S17c). Overall, in a real cellular environment, the riboswitch may quickly adjust its conformation with the change of surrounding ionic conditions, so as to facilitate ligand binding and regulate the expression of proteins.

Furthermore, at different ionic conditions, the interaction patterns between the residues in the three-way junction region of AARA are not exactly the same, resulting in the discrepancies of geometrical shape and internal space of the binding pocket. For example, in the presence of $Mg^{2+}$, residue U74 may directly interact with U51 by forming relatively stable hydrogen bond interactions, thereby compressing the space of binding pocket. However, the ligand only has a chance to bind to the AD in a pocket with the suitable shape and space. Consequently, in the real ionic environment, the highly flexible pocket region may need structural reorganization to adapt to the binding of ligand. As discussed in our previous work exploring the binding process of ligand to AARA under the condition of 150 mM $MgCl_2$, it is the swing out of U51 base that makes ligand enter into the binding pocket smoothly[46]. Even in the environments containing $Mg^{2+}$ ions, ligands still seek access to the binding pocket of AARA and eventually form more stable hydrogen bonding network and base stacking pattern to ensure the execution of regulatory function. Altogether, the influence of the coupling between metal ions and pocket structure on the ligand binding is a topic worthy of in-depth investigation. One thing to emphasize is that simulating the binding process of free ligands to aptamer is quite time-consuming. Despite this, it may be a suitable way to solve the problem by using enhanced sampling algorithms, such as Replica Exchange Molecular Dynamics (REMD)[60], Gaussian accelerated Molecular Dynamics (GaMD)[61], and so on.

Our work involve many cases of purely monovalent ionic environments, where it is quite difficult to experimentally capture the structure of binding pocket and obtain one-dimensional free energy profiles. Even so, some results are still meaningful to be verified using experimental techniques. In fact, single-molecule FRET experiments work well for the detection of biological phenomena in the distance scale of 1–10 nm[36,39]. Given that the criterion we used, $d_{U20-A23}$, may be less than 1 nm and therefore does not satisfy the condition for fluorescence detection, while the distance between the pocket-P1 (which is calibrated by residues A23 and U82 due to the instability of the P1 terminal base pair C13-G83, about 2–4 nm) may be suited for detection by applying fluorescence resonance effect (our statistical results are shown in Supplementary Fig. S18). Through the detection of probability distribution of $d_{A23-U82}$, it is possible to indirectly restore the free energy profiles of the structural transition of pocket region in different ionic environments.

In summary, the configuration of AARA can be controlled by the metal ion environment before ligand binding, and this change in configuration also has an essential impact on subsequent binding of ligands and the exercise of regulatory functions. Moreover, our present work provides valuable insights and effective analytical models for understanding the correlations between metal ion and regulatory mechanism of purine riboswitches.

## Methods

**System construction and molecular dynamics simulations**. In this study, four systems were designed to investigate the effects of metal ions on the structure of AARA (with and without ligand), which are listed as follow:

(1) $K_{neu\_free}$: AARA without ligand, neutralized by 70 $K^+$ counterions,
(2) $K_{neu\_bound}$: AARA with ligand, neutralized by 70 $K^+$ counterions,
(3) $K_{0.3\_free}$: AARA without ligand, neutralized by 70 $K^+$ counterions, plus 300 mM KCl,
(4) $Mg_{0.3\_free}$: AARA without ligand, neutralized by 70 $K^+$ counterions, plus 300 mM $MgCl_2$.

The initial atomic coordinates of AARA (as the initial structure for ligand-bound system) was taken from the X-ray experiment (Protein Data Bank ID: 1Y26)[28], and the ligand was removed to obtain an initial structure for ligand-free system. Each system was immersed in a truncated octahedron box filled with TIP3P[62] water molecules (at least a 12 Å buffer distance between the solute and edge of the periodic box). $K^+$ ions were placed randomly in the simulation box. To avoid the direct interactions between $Mg^{2+}$ and the AARA during the minimization and equilibration phases, $Mg^{2+}$ ions were placed at least 5 Å away from the RNA structure. All MD simulations were performed using the Assisted Model Building with Energy Refinement (AMBER) 18 software package[63,64] on NVIDIA GeForce RTX 2080 graphics cards. The parameters for $K^+$ and $Cl^-$ ions were derived from the previous work by ref. [65], and the parameters for $Mg^{2+}$ ions were provided by ref. [66,67], respectively. The parameters used for the RNA structure were AMBER ff99bsc0+χOL3 force field[68,69], and the general AMBER force field (GAFF)[70] combined with restrained electrostatic potential (RESP) charge calculation approach[71] was used to build the parameters of adenine ligand. Each system was energy minimized using the conjugate gradient method for 6000 steps. Then, using the Langevin thermostat[72], systems were heated from 0 to 300 K in 400 ps using position restraints with a force constant of 1000 kcal·mol⁻¹·Å⁻² to the RNA structure (NVT ensemble, T = 300 K). Subsequently, each system was gradually released in 5 ns (spending 1 ns each with position restraints of 1000, 100, 10, 1, and 0 kcal·mol⁻¹·Å⁻²) using the NPT ensemble (P = 1 bar, T = 300 K) before a production run. After that, the final structure of each system was subjected to two independent 1.2 μs MD simulations at constant temperature (300 K) and pressure (1 bar) with periodic boundary conditions and the particle mesh Ewald (PME) method[73]. In order to control constant pressure, we used the isotropic Berendsen barostat[74] with a time constant of 2 ps. During the equilibration and production process, the RNA structure was completely free in the solutions. Simulations were run with an integration step of 2 fs, and bond lengths for hydrogen atoms were fixed using the SHAKE algorithm[75]. PME electrostatics was calculated with an Ewald radius of 10 Å and the cutoff distance was also set to 10 Å for the van der Waals potential.

**Umbrella sampling**. The reaction coordinate (distance between residue U20 and residue A23) was divided into 25 windows spaced every 0.5 Å, which covered a range of 4 to 16 Å. The initial structures for each window were generated from the four original systems by steered MD simulations on AARA using a harmonic force constant of 5 kcal·mol⁻¹·Å⁻². The umbrella sampling for each window was carried out using the same protocol as the previous conventional MD simulations except for additional constraints with a harmonic force constant of 5 kcal·mol⁻¹·Å⁻², and each window was run for 100 ns, totaling 10 μs of simulation data for all four systems, where the first 20 ns of each simulation was removed from analysis for equilibration. Trajectories were analyzed using the weighted histogram analysis method (WHAM)[76] with implementation from the Grossfield group[77]. For the Monte Carlo bootstrapping error analysis, 100 trials were run for each distribution and the statistical inefficiency was calculated to obtain the number of statistically independent data points in each window. Convergence of each potential of mean force curve and normalized distributions of all sampling windows can be found in Supplementary Figs. S1, S2.

**Data extraction and molecular visualization**. Trajectories were processed and analyzed by the built-in Cpptraj module of Amber Tools package[78]. In order to ensure a rigorous comparison between different trajectories, we first removed the translational and rotational motion of AARA molecule. After that, a series of analyses were carried out on the trajectories, such as root-mean-square deviation (RMSD), root-mean-square fluctuation (RMSF), radius of gyration ($R_G$), hydrogen bond calculation, solvent-accessible surface area (SASA), principal component analysis (PCA), and so on. In the calculation of SASA, the solvent probe radius was set to 1.4 Å. It is worth noting that the result of SASA may be a negative value if only a small fraction of the solute is selected, and this is because the surface area calculated reflects the contribution of atoms in the selected part to the overall surface area of solute in the calculations of SASA. The helical parameters (such as central axis, H-rise, Opening, and so on) of three stems (P1, P2, and P3) of AARA are obtained using the program Curves+[79]. All the visualizations of molecular simulations were done by the Visual Molecular Dynamics (VMD) program[80]. It should be specified that the position of each residue (or phosphate group) of AARA is replaced by the phosphorus atom (where the negative charge is concentrated) within it, if not otherwise stated.

**Calculation of electrostatic potential surface**. The electrostatic potential surface of the AARA molecule was calculated by the Adaptive Poisson–Boltzmann Solver (APBS) software package[81]. The parameters employed for APBS are listed as follows: the system temperature is set to 300 K, the grid spacing is 0.3 Å, the dielectric

constants are 2.0 for AARA and 78.54 for water, and the solvent probe radius is 1.4 Å. The surface electrostatic potentials are visualized with the VMD program.

**Estimation of ligand binding free energy**. The free energies of ligand binding to the AARA were estimated by using MM-PBSA method[82] integrated in AMBER Tools. The binding free energies were computed by the following equation:

$$\Delta G_{bind} = \Delta G_{vdW} + \Delta G_{ele} + \Delta G_{pol} + \Delta G_{nonpol} - T\Delta S, \quad (1)$$

here, each term can be estimated by the energy of complex minus the sum of energies of aptamer and ligand. $\Delta G_{vdW}$ and $\Delta G_{ele}$ represent the van der Waals and electrostatic energies in the gas phase, and these two terms were computed based on the parameters set used in the MD simulations. $\Delta G_{pol}$ and $\Delta G_{nonpol}$ are polar and non-polar components of the solvation free energy. $\Delta G_{pol}$ was estimated by numerically solving the Poisson–Boltzmann equation, and $\Delta G_{nonpol}$ was simply obtained using an empirical equation $\Delta G_{nonpol} = \gamma SASA + \beta$, where SASA represent the solvent-accessible surface area. The surface tension $\gamma$ and the correction term $\beta$ were set to be 0.00542 kcal·mol⁻¹·Å⁻² and 0.92 kcal·mol⁻¹, respectively, in the AMBER package[83]. The translational and rotational entropies can be calculated using standard statistical mechanical formulas[84], and the vibrational entropy can be estimated using the normal-mode analysis[85]. Normal-mode analysis requires minimizing each frame, building the mass-weighted Hessian matrix, and diagonalizing it to obtain the vibrational frequencies $\nu_i$ (eigenvalues). Then, for a system containing $N$ atoms, the vibrational entropy can be given by[86]

$$S_{vib} = T^{-1} \sum_{i=1}^{3N-6} \left[ \frac{h\nu_i}{e^{h\nu_i/k_B T} - 1} - k_B T \ln\left(1 - e^{-h\nu_i/k_B T}\right) \right], \quad (2)$$

where $h$ is Planck's constant, $T$ is absolute temperature in Kelvin and $k_B$ is the Boltzmann's constant. Because normal-mode calculations are computationally demanding for large systems, we selected 100 evenly spaced snapshots along the whole production trajectories (in $K_{neu\_bound}$ system) for entropy estimations.

**Evaluation of global deformation for AARA structure**. To evaluate the global deformation of AARA at different ionic conditions, we calculated the change ratio $r_{\Delta d}(P_i, P_j)$ of the average distance between every two phosphate groups relative to that in crystal structure, and $r_{\Delta d}(P_i, P_j)$ is given by

$$r_{\Delta d}(P_i, P_j) = \frac{1}{T} \sum_{t=0}^{T} \frac{d_{P_i-P_j}(t) - d_{P_i-P_j}(Crys.)}{d_{P_i-P_j}(Crys.)}, \quad (3)$$

where $dP_i\text{-}P_j(t)$ denotes the time evolution of distance between phosphate group $i$ and $j$. $dP_i\text{-}P_j(Crys.)$ represents the distance between phosphate groups $i$ and $j$ in the crystal structure, and $T$ is the total simulation time. In addition, the standard deviation of $r_{\Delta d}(P_i, P_j)$ can give more information about the fluctuation range between phosphate groups.

**Measurement of electrostatic potential in different regions of AARA structure**. To clarify the effect of changing ionic environment in our simulations, we introduce a quantity to measure the electrostatic potential of any region on AARA structure, which is called the cumulative electrostatic potential strength coefficient $\varphi C$:

$$\varphi_C = \sum_{i=1}^{n} \frac{Z}{\| R_i - R_P \|}, \quad (4)$$

where $Z$ represents the valence of the group type we want to investigate, $R_P$ denotes the location of any region on AARA structure (simply marked by phosphate group), and $R_i$ represents the location of all groups of the type we focus on. In particular, for each phosphate group, the group itself at this position is not included in the summation.

**Statistics and reproducibility**. MD simulations were performed for each ionic condition using two replicas, and each simulation used the different random seeds to generate the initial velocity. No data was excluded from the sampled configurations, and no blinding methods were used in data analysis.

## Data availability

The simulation trajectory data that support the findings of this study are not openly available due to its huge size and are available from the corresponding author upon reasonable request. The initial and final PDB structure files of the full system and simulation input files can be obtained from Supplementary Data 1, and all source data underlying the graphs and charts presented in the main figures are provided in Supplementary Data 2.

## Code availability

All MD simulation trajectories were generated and analyzed using the Amber18 software package https://ambermd.org/. Molecular visualization were performed using the VMD software https://www.ks.uiuc.edu/Research/vmd/. C++ and python3 codes for simulation system initialization and data analysis are available for download at https://github.com/bolly-biophysics/biomolecule_dynamics.

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

## Acknowledgements

We thank Prof. Yun-Xing Wang for valuable discussions and Dr. Xi Zhang for critical reading of the manuscript. Support for this research was provided by the Cultivating Project for Young Scholar at Hubei University of Medicine [2020QDJZR015]; Advantages Discipline Group (Public health) Project in Higher Education of Hubei Province (2021-2025) [2022PHXKQ5]; National Natural Science Foundation of China [11874162].

## Author contributions

L.B. and Y.X. conceived the project. L.B. and W.B.K. performed simulations, developed and applied analysis algorithms. L.B. wrote the paper with supports from all authors.

## Competing interests

The authors declare no competing interests.
