## [Peer Review File · Communications Biology]

Reviewers' comments:

Reviewer #1 (Remarks to the Author):

This is a quite impressive computational study of effects of metal ion on riboswitch. The authors took add adenine riboswitch as an example to investigate the influences of metal ions on the structure and dynamics of riboswitch aptamer by using conventional molecular dynamic (cMD) simulations. The cMD simulation is on the micro second time scale, long enough to sample most probable conformation. The authors conclude that the two-state transition is not only related to the binding of cognate ligands, but also strongly coupled with the change of metal ion environments. The computational study utilized many state-of-the-art technologies. The quality of the paper is high. However, I have the following comments and questions that need to be addressed before the manuscript is published.

1. The authors placed the Mg²⁺ ions at least 5Å to RNA. Then, how to choose the sites of Mg²⁺? The local conformation of RNA without would be different from with Mg²⁺. In the equilibrated phase, the local conformation of RNA maybe changed. Then in production phase, how to fix the influences in the equilibrated phase.
2. We know that Mg²⁺ is particularly difficult to treat in simulations. While duplicating all of the MD simulations with a different ion model maybe a tall order, checking some of the conclusions against a reasonable alternative to confirm robustness or citing existed work (eg: Hu Guodong; Huan-Xiang Zhou, PLoS computational biology), is not too difficult.
3. The change ratio, $r\Delta d(P_i, P_j)$, is used to evaluate the global deformation. The change ratio reflects not only the relative change of two phosphate groups, but also the fluctuation of the phosphate group. For an example, a phosphate group fluctuates largely, its change ration with any one phosphate group would have a large value, but its average conformation would change a little. Then, how to get rid of the effects of this kind of fluctuation?
4. In the S1 section, Determination of the π - π stacking between two bases. The distance d_{ij} also needs a cutoff. In table S2, the data are averaged or from one snapshot in the traj. 1 and traj. 2?
5. The RMSFs can refelcts the fluctuation of RNA. Why the same systems show differet RMSFs? Such as, the RMSFs of Kneu_free1 is clearly larger than Kneu_free2 in Fig.8b, what happen? The robust of computational conclusion needs to be tested by the experiment. Then, it should be possible to formulate specific, experimentally testable predictions.

Reviewer #2 (Remarks to the Author):

In this work, standard MD simulation methods have been applied to study the effects of metal ions on the conformational profiles and dynamics of riboswitch in the free and ligand-bound form. Although the simulation and analysis appear to have been done with care, the connection with experimental evidence in relation to the findings from the simulation work presented here is not adequately addressed. I feel that the paper may be more suitable for journals specialized in molecular modeling.

1. The writing can be improved. For example, Page 2, line 37: "to clarify the conformational transition law of riboswitch in different external environments ..." may be better written as "to elucidate the nature of conformational transition in different environments..."
2. Page 12, Fig. 3 and related texts. How do the differences in the 1D conformational free energy profiles compare with experiments? More discussions that address the connections with the experimental evidence will certainly improve the manuscript.
3. In the MMPBSA calculation of ligand binding free energy, it is unclear if the one-trajectory or separate trajectory approach were used? How do these results compare with the experiments? In MMPBSA, the solute entropy change from the normal mode approximation Eq. 2. This approach has recently been shown to be the least accurate among different methods: see Wickstrom et al. Phys. Chem. Chem. Phys. 24, 6037-6052, 2022.
4. The manuscript should be significantly shortened, which will allow the main findings be highlighted.

According to the reviewers' suggestions and comments, we have carefully revised the manuscript. In addition, we have added 2 references (Refs. 82,85) and 2 figures (Supplementary Fig. S4 and Fig. S18) to enhance the manuscript. We use Red Color to highlight the localized changes we have made throughout the manuscript.

For reviewer #1

1. *“The authors placed the Mg^{2+} ions at least 5 Å to RNA. Then, how to choose the sites of Mg^{2+} ? The local conformation of RNA without would be different from with Mg^{2+} . In the equilibrated phase, the local conformation of RNA maybe changed. Then in production phase, how to fix the influences in the equilibrated phase.”*

Thank you for your nice comments, they are all very critical. In this work, we have written a C++ code to assign the positions of Mg^{2+} ions and placed them in a spherical layer 5 Å away from the surface of RNA molecules. The way to obtain the code has been placed in Code availability statement behind the main text. In addition, at the equilibrium stage, Mg^{2+} ions first completely were coordinated with water molecules. At the same time, the structure of RNA was constrained during most of the equilibrium simulation. RNA only interacts freely with Mg^{2+} ions in the last nanosecond, so the influence of equilibrium stage on the whole production phase is very subtle.

2. *“We know that Mg^{2+} is particularly difficult to treat in simulations. While duplicating all of the MD simulations with a different ion model maybe a tall order, checking some of the conclusions against a reasonable alternative to confirm robustness or citing existed work (eg: Hu Guodong; Huan-Xiang Zhou, PLoS computational biology), is not too difficult.”*

Thank you very much for your useful suggestion. To address the issue, we have added the following sentences in main text.

“In fact, a recent rigorous MD work showed a good agreement between the results of several currently popular force field combinations of RNA and Mg^{2+} ion for small riboswitches (82).” (page 29, lines 727–729 in the section of Discussion)

And one related paper are now cited,

82. Hu, G. and Zhou, H. X. (2021) Binding free energy decomposition and multiple unbinding paths of buried ligands in a PreQ₁ riboswitch. *PLoS Comput. Biol.*, **17**, e1009603.

3. “The change ratio, $r_{\Delta d}(P_i, P_j)$, is used to evaluate the global deformation. The change ratio reflects not only the relative change of two phosphate groups, but also the fluctuation of the phosphate group. For an example, a phosphate group fluctuates largely, its change ration with any one phosphate group would have a large value, but its average conformation would change a little. Then, how to get rid of the effects of this kind of fluctuation?”

Thank you very much for this meaningful question. To address the issue, we calculated the standard deviation of $r_{\Delta d}(P_i, P_j)$, which indeed contains more information. For example, there are large structural fluctuations near area C_{J12}-P₁ in K_{0.3_free_2} and K_{neu_bound_2}, which are caused by the two-state transition of C_{J12}-P₁ structure. Compared with the average value of $r_{\Delta d}(P_i, P_j)$, the effect of two-state transition is more obvious in the heat map of standard deviation. Moreover, we used the relative change ratio rather than the absolute distance between residues to measure the structural change properties, precisely to eliminate the impact of the benchmark difference in the distance between residues.

Accordingly, we have added the following sentences in the main text and Supplementary Fig. S4 in the supplementary information.

“The heat map of standard deviation of $r_{\Delta d}(P_i, P_j)$ strongly confirms the above facts. (Supplementary Fig. S4).” (page 12, lines 298–299 in the section of Results)

Fig. S4 The standard deviation of $r_{\Delta d}(P_i, P_j)$ for all eight simulation trajectories. The color bar shows the variations in standard deviation of $r_{\Delta d}(P_i, P_j)$ from 0 (dark blue) to 0.4 (red).

4. “In the S1 section, Determination of the π - π stacking between two bases. The

distance d_{ij} also needs a cutoff. In table S2, the data are averaged or from one snapshot in the traj. 1 and traj. 2?”

Thank you very much for your kind tips and questions, and we are sorry for this oversight. In this work, we set the cutoff of d_{ij} to 4.0 Å, a value slightly larger than the stacking distance between aromatic rings (~3.4 Å), which allows for small slips in the aromatic ring stacking. In addition, the data in table S2 are averaged from the traj. 1 and traj. 2.

Accordingly, we have added the following sentences in the supplementary information.

“(cutoff of $\|\mathbf{d}_{ij}\| = 4.0 \text{ \AA}$)” (page 2, line 33 in the section of Supplementary Methods) and “(average value)” (page 26, line 222 in the section of Supplementary Tables)

5. *“The RMSFs can reflect the fluctuation of RNA. Why the same systems show different RMSFs? Such as, the RMSFs of $K_{neu_free_1}$ is clearly larger than $K_{neu_free_2}$ in Fig. 8b, what happen?”*

Thank you for your nice comment. In fact, the RNA molecule is a highly flexible structure (especially in low concentration monovalent ionic environment), so even with simulation times of the order of microseconds, cMD may still under-sample the RNA, which leads to perceptible differences in the RMSFs of the same systems. As can be seen, the larger differences in RMSF are also mainly concentrated in the loop regions near the less stable P1 and binding pocket (especially the lack of ligand constraints). It is believed that if the simulation time is extremely long (tens of microseconds), the RMSFs of the same systems will definitely converge on a curve with subtle differences.

6. *“The robust of computational conclusion needs to be tested by the experiment. Then, it should be possible to formulate specific, experimentally testable predictions.”*

Thank you very much for your useful suggestion. Three of the four systems we simulated all contain only monovalent ions, and under these conditions, the RNA molecules are extremely flexible and X-ray diffraction methods cannot obtain sufficient lattice diffraction intensity, making it quite difficult to determine the structure of RNA. However, single-molecule FRET experiments work well for the detection of biological phenomena in the distance scale of 1–10 nm. Given that the

criterion we used, $d_{U20-A23}$, may be less than 1 nm and therefore does not satisfy the condition for fluorescence detection, while the distance between the pocket-P1 (which is calibrated by residues A23 and U82 due to the instability of the P1 terminal base pair C13-G83, about 2–4 nm) may be suited for detection by applying fluorescence resonance effect (our statistical results are shown in Supplementary Fig. S18). Therefore, it is expected that a follow-up experimental group will be able to validate our results in monovalent ionic environments, which are helpful in understanding the deformation property in pocket region and the subsequent regulatory mechanism of this family of riboswitches.

Accordingly, we have added the following sentences in the main text and Supplementary Fig. S18 in the supplementary information.

“Our work involve many cases of purely monovalent ionic environments, where it is quite difficult to experimentally capture the structure of binding pocket and obtain one-dimensional free energy profiles. Even so, some results are still meaningful to be verified using experimental techniques. In fact, single-molecule FRET experiments work well for the detection of biological phenomena in the distance scale of 1-10 nm (37,40). Given that the criterion we used, $d_{U20-A23}$, may be less than 1 nm and therefore does not satisfy the condition for fluorescence detection, while the distance between the pocket-P1 (which is calibrated by residues A23 and U82 due to the instability of the P1 terminal base pair C13-G83, about 2–4 nm) may be suited for detection by applying fluorescence resonance effect (our statistical results are shown in Supplementary Fig. S18). Through the detection of probability distribution of $d_{A23-U82}$, it is possible to indirectly restore the free energy profiles of the structural transition of pocket region in different ionic environments.” (page 32, lines 798–809 in the section of Discussion)

Fig. S18 **a** Schematic diagram of single-molecule FRET experiment design, the distance between A23 and U82 (filled with light green) may be suited for detection by applying fluorescence resonance effect. **b** The normalized distributions of $d_{A23-U82}$ for all eight simulation trajectories, and the gray dashed line corresponds to $d_{A23-U82}$ in the crystal structure (~ 29.3 Å).

For reviewer #2

1. “The writing can be improved. For example, Page 2, line 37: “to clarify the conformational transition law of riboswitch in different external environments ...” may be better written as “to elucidate the nature of conformational transition in different environments...””

Thank you very much for your good suggestion. To address the issue, we have rewritten this sentence (page 2, lines 36–37). In addition, we have proofread and polished the manuscript very carefully.

2. “Page 12, Fig. 3 and related texts. How do the differences in the 1D conformational free energy profiles compare with experiments? More discussions that address the connections with the experimental evidence will certainly improve the manuscript.”

Thank you very much for your useful suggestion. The 1D conformational free energy profiles are closely related to the structural distribution probability of riboswitches in solution. However, three of the four systems we simulated all contain only monovalent ions, and under these conditions, the RNA molecules are extremely flexible and X-ray diffraction methods cannot obtain sufficient lattice diffraction intensity, making it quite difficult to determine the structure of RNA. In the absence

of ligands and Mg^{2+} ions, the fine pocket structure is difficult to detect due to its variable structure. Therefore, no relevant experiments are currently available that give 1D free energy profiles for structural changes in the binding pocket. Nevertheless, assessing the effect of ionic environment or ligand binding on the structure of these regions is important for understanding the regulatory mechanisms of riboswitches. In fact, single-molecule FRET experiments work well for the detection of biological phenomena in the distance scale of 1–10 nm. Given that the criterion we used, $d_{U20-A23}$, may be less than 1 nm and therefore does not satisfy the condition for fluorescence detection, while the distance between the pocket-P1 (which is calibrated by residues A23 and U82 due to the instability of the P1 terminal base pair C13-G83, about 2–4 nm) may be suited for detection by applying fluorescence resonance effect (our statistical results are shown in Supplementary Fig. S18). Through the detection of probability distribution of $d_{A23-U82}$, it is possible to indirectly restore the 1D free energy profiles of the structural transition of pocket region in different ionic environments. Therefore, it is expected that a follow-up experimental group will be able to validate our results in monovalent ionic environments, which are helpful in understanding the deformation property in pocket region and the subsequent regulatory mechanism of this family of riboswitches.

Accordingly, we have added the following sentences in the main text and Supplementary Fig. S18 in the supplementary information.

“Our work involve many cases of purely monovalent ionic environments, where it is quite difficult to experimentally capture the structure of binding pocket and obtain one-dimensional free energy profiles. Even so, some results are still meaningful to be verified using experimental techniques. In fact, single-molecule FRET experiments work well for the detection of biological phenomena in the distance scale of 1-10 nm (37,40). Given that the criterion we used, $d_{U20-A23}$, may be less than 1 nm and therefore does not satisfy the condition for fluorescence detection, while the distance between the pocket-P1 (which is calibrated by residues A23 and U82 due to the instability of the P1 terminal base pair C13-G83, about 2–4 nm) may be suited for detection by applying fluorescence resonance effect (our statistical results are shown in Supplementary Fig. S18). Through the detection of probability distribution of $d_{A23-U82}$, it is possible to indirectly restore the free energy profiles of the structural transition of pocket region in different ionic environments.” (page 32, lines 798–809 in the section of Discussion)

Fig. S18 **a** Schematic diagram of single-molecule FRET experiment design, the distance between A23 and U82 (filled with light green) may be suited for detection by applying fluorescence resonance effect. **b** The normalized distributions of $d_{A23-U82}$ for all eight simulation trajectories, and the gray dashed line corresponds to $d_{A23-U82}$ in the crystal structure (~ 29.3 Å).

3. “In the MMPBSA calculation of ligand binding free energy, it is unclear if the one-trajectory or separate trajectory approach were used? How do these results compare with the experiments? In MMPBSA, the solute entropy change from the normal mode approximation Eq. 2. This approach has recently been shown to be the least accurate among different methods: see Wickstrom et al. *Phys. Chem. Chem. Phys.* 24, 6037-6052, 2022.”

Thank you very much for your good suggestion. In the MMPBSA calculation of ligand binding free energy, we used one-trajectory approach. Our calculation results are mainly compared with the previous work by Batey et al (*J. Mol. Biol.*, 359, 754–768, 2006). Their analysis revealed a highly favorable enthalpy of reaction at 30°C of -34.5 kcal·mol⁻¹ offset by a large and unfavorable entropy of reaction (-25.7 kcal·mol⁻¹). The solute entropy change from the normal mode approximation is ~ -18 kcal·mol⁻¹, which is somewhat different from the experimental results. Based on your suggestions, we have estimated conformational entropy using quasi-harmonic models in cartesian coordinates (*Phys. Chem. Chem. Phys.*, 24, 6037–6052, 2022), and obtained the entropy of $K_{neu_bound_1}$ and $K_{neu_bound_2}$ were -25.1 kcal·mol⁻¹ and -25.4 kcal·mol⁻¹, respectively, showing a good agreement with the experimental results (*J. Mol. Biol.*, 359, 754–768, 2006). Nevertheless, since the enthalpy contribution was also underestimated in MMPBSA calculation, the free energy obtained based on normal mode approximation did not deviate much from the

experiments. Given that the contribution of entropy is almost identical for both $K_{\text{neu_bound}}$ systems regardless of the method used, so it does not affect our conclusion essentially, that is, the more compact the structure of C_{J12-P1}, the tighter the ligand binding.

Correspondingly, we have added the following sentences in main text.

“In general, accurate estimation of entropy in MD simulation is a difficult task. For the sake of rigor, we estimated conformational entropy using quasi-harmonic models in cartesian coordinates (85), and obtained the entropy of $K_{\text{neu_bound_1}}$ and $K_{\text{neu_bound_2}}$ are $-25.1 \text{ kcal}\cdot\text{mol}^{-1}$ and $-25.4 \text{ kcal}\cdot\text{mol}^{-1}$, respectively, showing a good agreement with the experimental results (83). Nevertheless, since the enthalpy contribution was also underestimated in MMPBSA calculation, the free energy obtained based on normal mode approximation did not deviate much from the experiments. Given that the contribution of entropy is almost identical for both $K_{\text{neu_bound}}$ systems regardless of the method used, our results therefore clearly confirm that the ligand binds most tightly to the AARA only in the presence of Mg^{2+} ions (attributed to a more compact stacking pattern at the binding pocket).” (page 29–30, lines 733–743 in the section of Discussion)

And one related paper are now cited,

85. Wickstrom, L., Gallicchio, E., Chen, L., Kurtzman, T. and Deng, N. (2022) Developing end-point methods for absolute binding free energy calculation using the Boltzmann-quasiharmonic model. *Phys. Chem. Chem. Phys.*, **24**, 6037–6052.

4. *“The manuscript should be significantly shortened, which will allow the main findings be highlighted.”*

Thank you very much for your good suggestion. To address this issue, while maintaining the logical structure of the main text, we have deleted some unimportant statements, moved some main text to the Method section, and moved some pictures (original Fig. 7c & d) into the supplementary information (Supplementary Fig. S8), so as to try our best to improve the readability of the article.

Fig. 7 The structural changes around ligand binding pocket and their impacts on other regions of AARA. **a** The normalized distributions of R_G of binding pocket for all eight simulation trajectories, and the gray dashed line corresponds to R_G of binding pocket in the crystal structure (~ 10.6 Å). **b** The structure of ligand binding pocket in $Mg_{0.3_free}$ system (taking $Mg_{0.3_free_1}$ as an example), and the hydrogen bond interactions between important residues near binding pocket are indicated by red dashed lines. An important Mg^{2+} ion binding site (represented by green sphere) in this state is also marked in the figure. **c** A schematic diagram of change in base stacking structure caused by the bending deformation of the backbone at C_{J12-P1} . **d** The change of coaxial stacking morphology between P1 and P3 caused by the deformation of C_{J12-P1} . The central axes of P1 (red) and P3 (blue) are drawn with the van der Waals representation of hydrogen atom. The average values and standard deviations of Opening (**e**) and H-rise (**f**) of each base pair in P1 helix for all eight simulation trajectories, and the direction of P1 (from 5' end to binding pocket) is also shown in the figure.

Fig. S8 **a** The distances between two phosphate groups along the backbone in all eight simulation trajectories and crystal structure. **b** The bending angles composed by three consecutive adjacent phosphate groups along the backbone in all eight simulation trajectories and crystal structure.

REVIEWERS' COMMENTS:

Reviewer #2 (Remarks to the Author):

The responses and the revision of the manuscript are satisfactory. It can be accepted now.

Reviewer #3 (Remarks to the Author):

The revised manuscript has adequately addressed my previous questions/concerns and is substantially improved.